# ROBUST NON-NEGATIVE PROXIMAL GRADIENT ALGORITHM: THEORY AND APPLICATIONS

## ABSTRACT

Proximal gradient algorithms (PGA), while foundational for inverse problems like image reconstruction, often yield unstable convergence and suboptimal solutions by violating the critical non-negativity constraint. We identify the gradient descent step as the root cause of this issue, which introduces negative values and induces high sensitivity to hyperparameters. To overcome these limitations, we propose a novel multiplicative update proximal gradient algorithm (SSO-PGA) with convergence guarantees, which is designed for robustness in non-negative inverse problems. Our key innovation lies in superseding the gradient descent step with a learnable sigmoid-based operator, which inherently enforces non-negativity and boundedness by transforming traditional subtractive updates into multiplicative ones. This design, augmented by a sliding parameter for enhanced stability and convergence, not only improves robustness but also boosts expressive capacity and noise immunity. We further formulate a degradation model for multi-modal restoration and derive its SSO-PGA-based optimization algorithm, which is then unfolded into a deep network to marry the interpretability of optimization with the power of deep learning. Extensive numerical and real-world experiments demonstrate that our method significantly surpasses traditional PGA and other state-of-the-art algorithms, ensuring superior performance and stability.

## 1 INTRODUCTION

This paper focuses on the following convex optimization problems:

$$\min_x F(x), \quad \text{s.t. } x > 0, \quad \text{where} \quad \begin{cases} F(x) = f(x), & \text{(Problem I)}, \\ F(x) = f(x) + g(x), & \text{(Problem II)}. \end{cases} \tag{1}$$

Here, $f$ is a convex and differentiable function, while $g$ is a convex but not necessarily smooth function. For Problem I (unconstrained convex and differentiable problem), researchers commonly use the classic gradient descent method for a solution Ruder (2016). However, for Problem II (non-smooth composite optimization problem), which includes a non-differentiable term, researchers have explored various solution methods Li et al. (2021). The most common among these are splitting algorithms Goldfarb & Ma (2012), which use first-order information to minimize the objective function. These include: the proximal gradient algorithm (PGA) Li & Lin (2015); Salim et al. (2020), the alternating direction method of multipliers (ADMM) Boyd et al. (2011); Hong & Luo (2017), the Douglas-Rachford splitting (DRS) Eckstein & Bertsekas (1992); Patrinos et al. (2014), and the Pock-Chambolle (PC) algorithm Chambolle & Pock (2011). Among these, PGA is particularly popular due to its sound theoretical foundation and ease of optimization Dai et al. (2024).

The core idea of PGA is to perform a standard gradient descent step on $f$ followed by a proximal projection on $g$ Laude & Patrinos (2025). To accelerate convergence and enhance stability, researchers have introduced numerous improvements Li et al. (2019b); Iutzeler & Malick (2018); Si et al. (2024). For instance, Keys et al. proposed the proximal distance algorithm, which combined classical penalty methods with distance majorization techniques Keys et al. (2019). Additionally, Malitsky et al. introduced an adaptive proximal gradient method that leveraged the local curvature information of the smooth function to achieve full adaptivity Malitsky & Mishchenko (2024).

PGA provides a foundation for solving inverse problems in signal processing Antonello et al. (2018), compressed sensing Yao & Dai (2025), and image reconstruction Shen et al. (2011). With the rise of

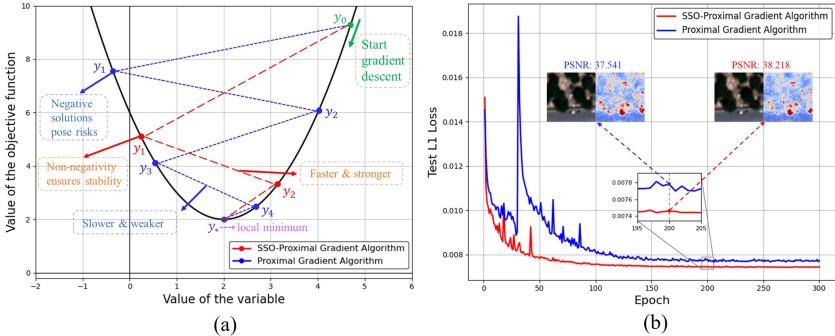

(a)           (b)

Figure 1: Overview of our method. (a) A schematic comparison between PGA and SSO-PGA in the gradient descent process. Compared to PGA, SSO-PGA benefits from the non-negativity constraint, yielding more stable solutions and demonstrating a faster convergence trajectory. (b) Comparison of test L1 loss curves between PGA and SSO-PGA on the WV3 dataset in the image fusion task over training epochs, with a zoomed-in view highlighting the reconstructed results at epoch 200. SSO-PGA exhibits a more stable training process and achieves superior fusion quality.

deep learning, PGA has been successfully integrated into deep unfolding networks, creating a hybrid paradigm that integrates iterative optimization with learnable components to boost performance Wei et al. (2022); Mou et al. (2022). This approach models the problem to be solved as an optimization objective and uses deep priors as the function $g$. In this area, Mardani et al. first proposed a novel neural proximal gradient descent algorithm that uses a recurrent ResNet to learn the proximal mapping, enabling high-resolution image recovery from limited sensory data Mardani et al. (2018). Xin et al. further improved deep unfolding networks by introducing an adaptive learning rate and borrowing the momentum technique from gradient descent, proposing a multi-stage and multi-level feature aggregation scheme for efficient MRI reconstruction Xin et al. (2024).

Despite the significant achievements of deep unfolding algorithms in vision tasks, their application still faces challenges. Their performance is often limited by the hyperparameter settings of the PGA, leading to unstable and suboptimal results. Furthermore, in vision tasks, images inherently have a non-negativity constraint. However, traditional PGA can produce negative solutions during the iterative process. Although these negative values may be numerically plausible, they violate the physical constraints of images and can exacerbate instability within the deep network during iteration.

To this end, we propose a novel robust non-negative proximal gradient algorithm (SSO-PGA), which maintains the optimization simplicity of the traditional PGA while effectively overcoming its drawbacks of instability and sensitivity. For Problem I, we reformulate the conventional additive gradient descent step into a new multiplicative update scheme via a Sliding Sigmoid Operator (SSO). Unlike traditional sensitive step sizes that often cause overshooting or vanishing updates, SSO adapts dynamically to the local gradient landscape, allowing finer control over the descent direction and magnitude. This leads to smoother convergence and mitigates abrupt changes. For Problem II, we can naturally extend the gradient descent algorithm from Problem I to the proximal gradient algorithm (SSO-PGA) by adding a proximal projection. Moreover, the inherent non-linearity and non-negativity of the SSO-PGA enhance robustness to noise. These properties make SSO-PGA particularly well-suited for vision tasks, as images inherently possess non-negative physical constraints. *To our knowledge, this is the first work that improves upon PGA by using a multiplicative approach to fundamentally guarantee non-negativity and robustness, and adapt it to a deep network framework.* As shown in Fig. 1, compared with the existing PGA, SSO-PGA achieves more stable solutions, faster convergence, and superior performance, without introducing additional hyperparameters. The main contributions of this work are summarized as follows:

- We propose a novel robust non-negative proximal gradient algorithm (SSO-PGA) with theoretical convergence guarantees, which improves the gradient descent step of the traditional PGA via the Sliding Sigmoid Operator. This innovation inherently enforces non-negativity constraints, enhances nonlinear representation, and improves numerical stability.

- Based on the proposed SSO-PGA, we develop a novel inverse problems model with efficient optimization. Specifically, we formulate Problem II as a multi-modal restoration

problem and derive the corresponding optimization paradigm. This model is further unfolded into a structured deep neural network.

- Numerical experiments demonstrate superior performance for both Problem I and Problem II. Our deep unfolding network also shows a significant advantage in vision experiments, surpassing both the PGA baseline and other state-of-the-art (SOTA) algorithms for vision tasks. Moreover, compared to the PGA baseline, our SSO-PGA significantly improves convergence speed, hyperparameter stability, and robustness against perturbations.

## 2 RELATED WORK

Inverse problems are widespread across various fields, where one seeks to recover an unknown $\boldsymbol{y} \in \mathbb{R}^m$ from partial observations $\boldsymbol{x} \in \mathbb{R}^n$ Deng et al. (2018); Farahmand-Tabar et al. (2024). This is often based on a Gaussian noise assumption ($\boldsymbol{x} = \boldsymbol{H}\boldsymbol{y} + \boldsymbol{n}$) and can be represented as:

$$\min_{\boldsymbol{y}} \|\boldsymbol{x} - \boldsymbol{H}\boldsymbol{y}\|_2^2, \quad \text{(Problem I)}, \tag{2}$$

To achieve a more accurate recovery, researchers often introduce prior information Nan & Ji (2020):

$$\min_{\boldsymbol{y}} \|\boldsymbol{x} - \boldsymbol{H}\boldsymbol{y}\|_2^2 + \lambda f(\boldsymbol{y}), \quad \text{(Problem II)}, \tag{3}$$

where $\boldsymbol{H} \in \mathbb{R}^{n \times m}$ is a degradation operator, and $f(\boldsymbol{y})$ is a regularization term that encodes prior knowledge about $\boldsymbol{y}$. When $f(\boldsymbol{y})$ is convex but possibly non-smooth (e.g., $\ell_1$-norm He et al. (2014) or total variation Palsson et al. (2013)), the proximal gradient algorithm provides an efficient first-order method to solve the problem. Specifically, the update rule of the proximal gradient algorithm at the $t$-th iteration is given by Beck & Teboulle (2009):

$$\boldsymbol{y}^t = Prox_f\left(\boldsymbol{y}^{t-1} - \rho \nabla \mathcal{E}(\boldsymbol{y}^{t-1})\right), \quad \mathcal{E}(\boldsymbol{y}^{t-1}) = \|\boldsymbol{x} - \boldsymbol{H}\boldsymbol{y}^{t-1}\|_2^2, \tag{4}$$

where $\rho$ is a step size, and the proximal operator is defined as:

$$Prox_f(\boldsymbol{v}) = \arg\min_{\boldsymbol{z}} \left\{ \frac{1}{2} \|\boldsymbol{z} - \boldsymbol{v}\|_2^2 + \lambda f(\boldsymbol{z}) \right\}. \tag{5}$$

Although the proximal gradient algorithm enjoys fast convergence, it suffers from a major drawback: in imaging applications, pixel intensities are inherently non-negative, yet the update rule in Eq. (4) may yield negative values. This not only violates the natural characteristics of images but also introduces vanishing gradient issues when implemented in deep unfolding networks. A straightforward solution to this problem is to restrict the update step by setting the step size $\rho$ as follows Lee & Seung (2000):

$$\rho_i = \frac{\boldsymbol{y}_i^{t-1}}{(\boldsymbol{H}^\top \boldsymbol{H} \boldsymbol{y}^{t-1})_i}, \quad \text{for } i = 1, \ldots, m. \tag{6}$$

Substituting this into Eq. (4) yields the following update rule:

$$\begin{aligned}
\boldsymbol{y}_i^t &= Prox_f\left(\boldsymbol{y}_i^{t-1} - \frac{\boldsymbol{y}_i^{t-1}}{(\boldsymbol{H}^\top \boldsymbol{H} \boldsymbol{y}^{t-1})_i}\left((\boldsymbol{H}^\top \boldsymbol{H} \boldsymbol{y}^{t-1})_i - (\boldsymbol{H}^\top \boldsymbol{x})_i\right)\right) \\
&= Prox_f\left(\frac{\boldsymbol{y}_i^{t-1}}{(\boldsymbol{H}^\top \boldsymbol{H} \boldsymbol{y}^{t-1})_i}(\boldsymbol{H}^\top \boldsymbol{x})_i\right).
\end{aligned} \tag{7}$$

While this formulation guarantees non-negativity, it introduces a new numerical challenge: division by zero. Even when a small stabilization constant is introduced, this issue still results in numerical instability. This problem becomes more pronounced in deep unfolding networks, where it will prone to yield convergence failure or gradient explosion.

## 3 METHOD

### 3.1 SSO-ENHANCED PROXIMAL GRADIENT ALGORITHM

The motivation of this work is to address the non-negativity constraint in the proximal gradient algorithm while ensuring stability and robustness in both iterative optimization and deep learning frameworks. First, we give the definition of the Sliding Sigmoid Operator.

**Definition 1.** *We define the Sliding Sigmoid Operator (SSO) as follows:*

$$SSO_\alpha(z) = 2\sigma(-z - \alpha) + 2\sigma(\alpha) - 1, \tag{8}$$

*where $\sigma(c) = \frac{1}{1+e^{-c}}$ denotes the sigmoid function, and $\alpha$ is the sliding parameter.*

As shown in Fig. 2, SSO is essentially a sigmoid function augmented with a sliding parameter $\alpha$. Specifically, as $\alpha$ varies, the sigmoid curve slides along the coordinate point $(0, 1)$, adjusting its upper and lower bounds accordingly. Notably, the function always passes through the point $(0, 1)$, ensuring that its output is less than 1 when the input is positive, and greater than 1 when the input is negative. When the gradient is used as the input variable, this property, combined with the multiplicative update, naturally implements a gradient descent behavior. Furthermore, by adjusting $\alpha$, SSO adaptively controls the step size in the gradient descent process. Thereby, we can define the update rule of the SSO-enhanced proximal gradient algorithm in **Definition 2**:

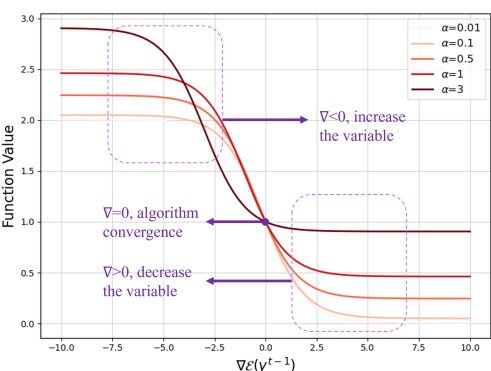

Figure 2: SSO working mechanism and its function curves under different $\alpha$ values.

**Definition 2.** *The update rule of the SSO-enhanced proximal gradient algorithm (SSO-PGA) to the inverse problem in Eq. (3) at the t-th iteration is defined as follows:*

$$\begin{aligned}
\boldsymbol{y}^t &= \boldsymbol{y}^{t-1} \odot SSO_\alpha(\nabla\mathcal{E}(\boldsymbol{y}^{t-1})), && \textit{(For Problem I)}, \\
\boldsymbol{y}^t &= Prox_f\left(\boldsymbol{y}^{t-1} \odot SSO_\alpha(\nabla\mathcal{E}(\boldsymbol{y}^{t-1}))\right), && \textit{(For Problem II)},
\end{aligned} \tag{9}$$

where $\odot$ denotes the element-wise product. Through SSO-PGA, we not only preserve the original gradient descent mechanism, but also constrain the updated variable within a multiplicative range of $(2\sigma(\alpha) - 1, 2\sigma(\alpha) + 1)$ relative to the original variable, thereby enabling more robust gradient descent. Moreover, the SSO multiplier enforces non-negativity of the updated variable, which better aligns with the characteristics of natural images. Then, we provide the following **Theorem 1**.

**Theorem 1.** *There exists $\rho_i > 0$ such that the SSO update rule is equivalent to a standard gradient descent step:*

$$\boldsymbol{y}_i^t = \boldsymbol{y}_i^{t-1} \cdot SSO_\alpha\left(\nabla\mathcal{E}(\boldsymbol{y}_i^{t-1})\right) = \boldsymbol{y}_i^{t-1} - \rho_i\nabla\mathcal{E}(\boldsymbol{y}_i^{t-1}), \quad for \ i = 1, \ldots, m. \tag{10}$$

Please refer to the *APPENDIX* for the proof. **Theorem 1** demonstrates that SSO-PGA retains the fundamental logic of traditional gradient descent. SSO-PGA integrates the nonlinear representational capacity of the Sliding Sigmoid Operator with the theoretical foundation of gradient descent, enabling it to maintain stability while offering greater flexibility for adaptive adjustment.

Here, we prove the convergence of SSO-PGA. As the proximal step is unchanged from PGA, we only prove the gradient descent part. First, we introduce three lemmas.

**Lemma 1.** *For every $\alpha \geq 0$ and every $z \in \mathbb{R}$, the following hold:*

$$\left|SSO_\alpha(z) - 1\right| \leq \eta(\alpha)\left|z\right|, \quad \eta(\alpha) = \frac{1+\alpha}{2}. \tag{11}$$

**Lemma 2.** *Let $\mathcal{E}: \mathbb{R}^n \to \mathbb{R}$ have $L$–Lipschitz gradient. Then for any $\boldsymbol{y}, \boldsymbol{d} \in \mathbb{R}^n$ Nesterov (2013):*

$$\mathcal{E}(\boldsymbol{y} + \boldsymbol{d}) \leq \mathcal{E}(\boldsymbol{y}) + \langle\nabla\mathcal{E}(\boldsymbol{y}), \boldsymbol{d}\rangle + \frac{L}{2}\left\|\boldsymbol{d}\right\|_2^2. \tag{12}$$

**Lemma 3.** *Given $\mathcal{E}(\boldsymbol{y}) = \|\boldsymbol{x} - \boldsymbol{H}\boldsymbol{y}\|_2^2$, for all $\boldsymbol{y}, \boldsymbol{z} \in \mathbb{R}^n$, the following hold:*

$$\left\|\nabla\mathcal{E}(\boldsymbol{y}) - \nabla\mathcal{E}(\boldsymbol{z})\right\|_2 \leq L\|\boldsymbol{y} - \boldsymbol{z}\|_2, \quad L = 2\|\boldsymbol{H}\|_2^2. \tag{13}$$

**Theorem 2.** *Let $0 \leq \alpha \leq 2/(\kappa\|\boldsymbol{H}\|_2^2) - 1$, the inverse problem $\|\boldsymbol{x} - \boldsymbol{H}\boldsymbol{y}\|_2^2$ is nonincreasing under the update rule:*

$$\boldsymbol{y}^t = \boldsymbol{y}^{t-1} \odot SSO_\alpha(\nabla\mathcal{E}(\boldsymbol{y}^{t-1})), \tag{14}$$

*where $\kappa = \|\boldsymbol{y}^{t-1}\|_\infty$.*

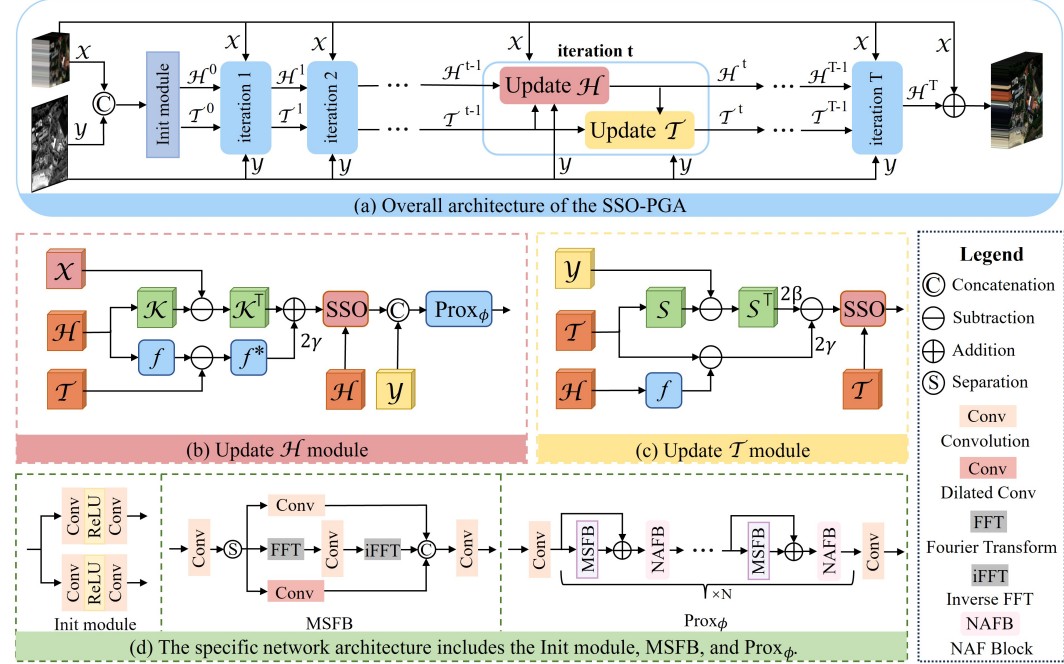

Figure 3: The network architecture of our method. (a) SSO-PGA consists of $T$ iterative steps, where each iteration includes (b) the update of $\mathcal{H}$ and (c) the update of $\mathcal{T}$. (d) The detailed network architecture of SSO-PGA, including the init module, MSFB block, and $Prox_\phi(\cdot)$ from left to right.

Please refer to the *APPENDIX* for the proof. From **Theorem 2**, combined with the fact that $\mathcal{E}(\boldsymbol{y}^t) \geq 0$ for every $t \geq 1$, we can conclude that the inverse problem $\|\boldsymbol{x} - \boldsymbol{H}\boldsymbol{y}\|_2^2$ converges to a local minimum under the gradient descent rule based on the SSO. It is worth noting that the condition $0 \leq \alpha \leq 2/(\kappa\|\boldsymbol{H}\|_2^2) - 1$, used in the proof is merely a sufficient condition for ease of analysis. In experiments, we have found that $\alpha$ admits a much broader range of values.

## 3.2 FORMULATION AND OPTIMIZATION

We formulate the SSO-PGA framework for solving inverse problems. Using multi-modal restoration as an example, given an observed image $\mathcal{X} \in \mathbb{R}^{h \times w \times C_1}$ and a guided image $\mathcal{Y} \in \mathbb{R}^{H \times W \times C_2}$, our goal is to reconstruct the target image $\mathcal{H} \in \mathbb{R}^{H \times W \times C_1}$. We explicitly model the degradation processes in both domains to capture the differences between different modalities:

$$\min_{\mathcal{H}, \mathcal{T}} \|\mathcal{X} - \mathcal{K}\mathcal{H}\|_F^2 + \beta\|\mathcal{Y} - \mathcal{S}\mathcal{T}\|_F^2, \tag{15}$$

where $\mathcal{T} \in \mathbb{R}^{H \times W \times C_2}$ denotes the guided-aligned latent embedding of the target image. $\mathcal{K}$ and $\mathcal{S}$ represent different degradation operators. We further enforce cross-domain consistency between the target image features and their guided-aligned embedding, thereby jointly preserving details in both domains:

$$\min_{\mathcal{H}, \mathcal{T}} \|\mathcal{X} - \mathcal{K}\mathcal{H}\|_F^2 + \beta\|\mathcal{Y} - \mathcal{S}\mathcal{T}\|_F^2 + \gamma\|\mathcal{T} - f(\mathcal{H})\|_F^2, \tag{16}$$

where $f(\cdot)$ is a feature transformation network. Finally, a deep prior $\phi(\cdot)$ is incorporated to further enhance the reconstruction quality of the target image. The final optimization objective can be formulated as:

$$\min_{\mathcal{H}, \mathcal{T}} \|\mathcal{X} - \mathcal{K}\mathcal{H}\|_F^2 + \beta\|\mathcal{Y} - \mathcal{S}\mathcal{T}\|_F^2 + \gamma\|\mathcal{T} - f(\mathcal{H})\|_F^2 + \phi(\mathcal{H}). \tag{17}$$

Based on the SSO-PGA, we update each variable alternately.

**Step 1:** $\mathcal{H}$ can be updated as follows at the t-th iteration:

$$\mathcal{H}^t = Prox_\phi\left(\mathcal{H}^{t-1} \odot SSO_{\alpha_1}\left(\nabla\mathcal{E}(\mathcal{H}^{t-1})\right)\right), \tag{18}$$

where

$$\mathcal{E}(\boldsymbol{\mathcal{H}}^{t-1}) = \|\boldsymbol{\mathcal{X}} - \boldsymbol{\mathcal{K}}\boldsymbol{\mathcal{H}}^{t-1}\|_F^2 + \gamma\|\boldsymbol{\mathcal{T}}^{t-1} - f(\boldsymbol{\mathcal{H}}^{t-1})\|_F^2, \tag{19}$$

and

$$\nabla\mathcal{E}(\boldsymbol{\mathcal{H}}^{t-1}) = 2\boldsymbol{\mathcal{K}}^\top(\boldsymbol{\mathcal{K}}\boldsymbol{\mathcal{H}}^{t-1} - \boldsymbol{\mathcal{X}}) + 2\gamma f^*\big(f(\boldsymbol{\mathcal{H}}^{t-1}) - \boldsymbol{\mathcal{T}}^{t-1}\big). \tag{20}$$

Specifically, $f^*(\cdot)$ is the subgradient of $f(\cdot)$, and the proximal operator $Prox_\phi(\cdot)$ is a deep network related to $\phi(\cdot)$.

**Step 2:** Similarly, we update $\boldsymbol{\mathcal{T}}$ as follows:

$$\boldsymbol{\mathcal{T}}^t = \boldsymbol{\mathcal{T}}^{t-1} \odot SSO_{\alpha_2}\big(\nabla\mathcal{E}(\boldsymbol{\mathcal{T}}^{t-1})\big), \tag{21}$$

where

$$\mathcal{E}(\boldsymbol{\mathcal{T}}^{t-1}) = \beta\|\boldsymbol{\mathcal{Y}} - \boldsymbol{\mathcal{S}}\boldsymbol{\mathcal{T}}^{t-1}\|_F^2 + \gamma\|\boldsymbol{\mathcal{T}}^{t-1} - f(\boldsymbol{\mathcal{H}}^t)\|_F^2, \tag{22}$$

and

$$\nabla\mathcal{E}(\boldsymbol{\mathcal{T}}^{t-1}) = 2\beta\boldsymbol{\mathcal{S}}^\top(\boldsymbol{\mathcal{S}}\boldsymbol{\mathcal{T}}^{t-1} - \boldsymbol{\mathcal{Y}}) + 2\gamma(\boldsymbol{\mathcal{T}}^{t-1} - f(\boldsymbol{\mathcal{H}}^t)). \tag{23}$$

### 3.3 Deep Unfolding Network

This subsection unfolds the SSO-PGA framework into a deep network architecture. As shown in Fig. 3, the network begins with an initialization module, followed by multiple iterative stages. Each iteration comprises two submodules: one for updating $\boldsymbol{\mathcal{H}}$ and the other for updating $\boldsymbol{\mathcal{T}}$. In this formulation, the operators $\boldsymbol{\mathcal{K}}, \boldsymbol{\mathcal{K}}^\top, \boldsymbol{\mathcal{S}}, \boldsymbol{\mathcal{S}}^\top$ in the original optimization steps are replaced by a multi-scale spatial frequency feature extraction module (MSFB), while the functions $f(\cdot)$ and $f^*(\cdot)$ are implemented using an NAFBlock Chen et al. (2022). The proximal operator $Prox_\phi(\cdot)$ is modeled by a combination of multiple MSFBs and NAFBlocks Chen et al. (2022). Additionally, all hyperparameters in each iteration, including $\beta$, $\gamma$, $\alpha_1$, and $\alpha_2$, are learnable and passed through a Softplus function to enforce non-negativity. Finally, the final network output is obtained by adding the $\boldsymbol{\mathcal{H}}$ in the last iteration and the initial input, and an L1 loss is applied against the ground truth.

## 4 Experiments

In this section, we conduct comprehensive experiments to validate the effectiveness of our method, including both numerical experiments and real-world vision experiments.

### 4.1 Comparison with Traditional Proximal Gradient Algorithm

#### 4.1.1 Numerical Experiments

In this subsection, we construct two convex optimization problems and perform numerical simulation experiments.

$$\min_y (y - 0.5)^2, \qquad \text{(Problem I)},$$
$$\min_y (y - 0.5)^2 + \frac{1}{2}|y|, \qquad \text{(Problem II)}. \tag{24}$$

We selected initial values of 1, 4, 8, and 16, with learning rates of 0.0005 and 0.005 (For additional experiments, please refer to the *APPENDIX*). Fig. 4 and Fig. 5 show that our SSO-PGA has a clear advantage over PGA, which can be attributed to the benefits of our multiplicative update rule.

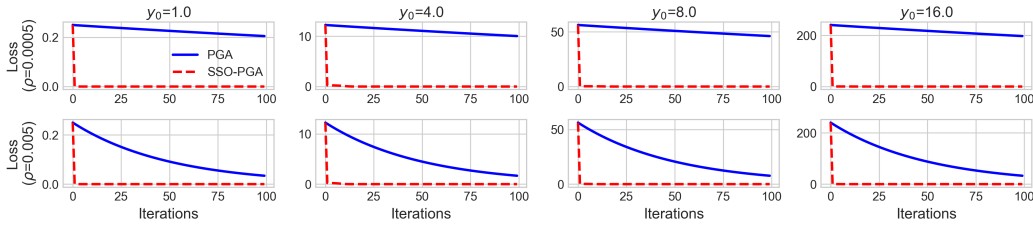

Figure 4: Comparison of numerical simulation results for SSO-PGA and PGA on Problem I.

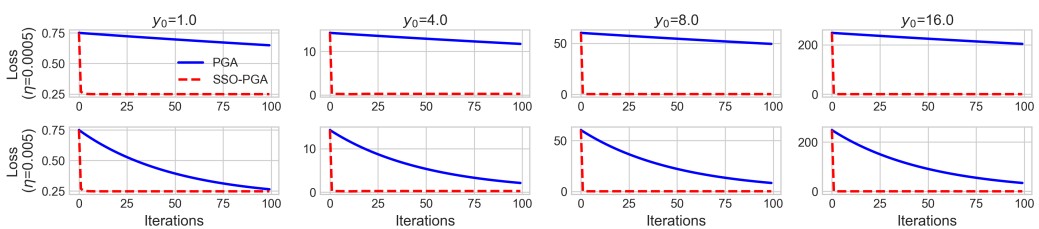

Figure 5: Comparison of numerical simulation results for SSO-PGA and PGA on Problem II.

### 4.1.2 REAL-WORLD VISION EXPERIMENTS

In this subsection, we construct a PGA baseline by replacing the SSO update rule in Eq. (9) with the traditional gradient descent formulation in Eq. (4), while keeping all other components unchanged. We then conduct a comprehensive comparison with our proposed SSO-PGA.

**Performance Comparison.** To more intuitively verify the effectiveness of SSO, in addition to comparing SSO-PGA with PGA, we also replace the traditional gradient descent step in MDCUN Yang et al. (2022) with our SSO-based update rule and compare it with the original version. As shown in Tab. 1, the SSO-enhanced models significantly outperform the traditional gradient descent models across all three datasets, demonstrating the superiority of the proposed SSO update mechanism.

Table 1: Quantitative comparison of traditional proximal gradient algorithm and SSO-enhanced proximal gradient algorithm on three datasets: WV3, QB, and GF2. The better results are in **bold**.

| Methods | WV3 | | | | QB | | | | GF2 | | | |
|---|---|---|---|---|---|---|---|---|---|---|---|---|
| | PSNR↑ | SAM↓ | ERGAS↓ | Q8↑ | PSNR↑ | SAM↓ | ERGAS↓ | Q4↑ | PSNR↑ | SAM↓ | ERGAS↓ | Q4↑ |
| MDCUN Yang et al. (2022) | 37.973 | 3.298 | 2.479 | **0.909** | 36.178 | **4.963** | 4.698 | 0.915 | 41.138 | 0.870 | 0.815 | 0.974 |
| SSO-MDCUN | **38.135** | **3.222** | **2.437** | **0.909** | **36.462** | 5.007 | **4.527** | **0.917** | **41.626** | **0.869** | **0.783** | **0.976** |
| PGA | 39.145 | 2.925 | 2.129 | 0.918 | 38.628 | 4.430 | 3.557 | 0.937 | 43.411 | 0.697 | 0.615 | 0.982 |
| SSO-PGA | **39.358** | **2.823** | **2.078** | **0.921** | **38.807** | **4.312** | **3.493** | **0.938** | **44.005** | **0.660** | **0.574** | **0.985** |

**Convergence Behavior.** Fig. 6 illustrates the convergence behavior of SSO-PGA and PGA under varying numbers of iterations. As observed, both methods perform comparably at the first iteration, because the model at this stage mainly behaves like a deep network, and the iterative formulation has not yet taken effect. However, with just two iterations, SSO-PGA already surpasses the three-iteration performance of PGA. By the third iteration, SSO-PGA exceeds the best performance achieved by PGA. Notably, at higher iteration counts, PGA exhibits signs of performance degradation, whereas SSO-PGA continues to improve steadily. This indicates that SSO-PGA not only converges faster but is also more robust against falling into poor local minima.

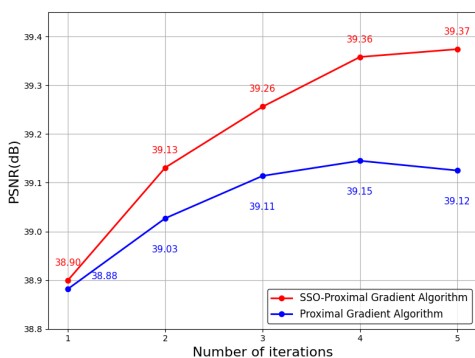

Figure 6: PSNR comparison of SSO-PGA and PGA over iterations on the WV3 dataset.

**Parameter Sensitivity.** Both SSO-PGA and PGA involve two hyperparameters during the update process: the sliding factor $\alpha_1, \alpha_2$ for SSO-PGA and the step size $\rho_1, \rho_2$ for PGA. As noted in our deep unfolding network, these hyperparameters are learnable. Here, we assign multiple initial values to $\alpha$ and $\rho$ to evaluate the sensitivity of SSO-PGA and PGA to the hyperparameter. Tab. 2 shows that PGA achieves its best performance when $\rho = 0.1$, and suboptimal results when $\rho = 0.01$. In contrast, SSO-PGA consistently performs well across all initial values. Notably, when the hyperparameter is set to relatively large values (e.g., $3.0$ or $5.0$), PGA fails to converge, whereas SSO-PGA still delivers strong performance. These results further confirm the robustness and stability of the proposed SSO-PGA framework.

Table 2: Quantitative comparison of SSO-PGA and PGA on the WV3 reduced-resolution dataset with varying parameter initialization settings. The better results are in **bold**.

| Parameter | $\alpha, \rho = 0.01$ | | | | $\alpha, \rho = 0.1$ | | | | $\alpha, \rho = 0.5$ | | | |
|---|---|---|---|---|---|---|---|---|---|---|---|---|
| | PSNR↑ | SAM↓ | ERGAS↓ | Q8↑ | PSNR↑ | SAM↓ | ERGAS↓ | Q8↑ | PSNR↑ | SAM↓ | ERGAS↓ | Q8↑ |
| PGA | 39.116 | 2.948 | 2.143 | 0.919 | 39.145 | 2.925 | 2.129 | 0.918 | 39.063 | 2.923 | 2.149 | 0.918 |
| SSO-PGA | **39.223** | **2.859** | **2.119** | **0.920** | **39.191** | **2.863** | **2.116** | **0.920** | **39.283** | **2.847** | **2.095** | **0.920** |

| Parameter | $\alpha, \rho = 1.0$ | | | | $\alpha, \rho = 3.0$ | | | | $\alpha, \rho = 5.0$ | | | |
|---|---|---|---|---|---|---|---|---|---|---|---|---|
| | PSNR↑ | SAM↓ | ERGAS↓ | Q8↑ | PSNR↑ | SAM↓ | ERGAS↓ | Q8↑ | PSNR↑ | SAM↓ | ERGAS↓ | Q8↑ |
| PGA | 38.907 | 3.017 | 2.195 | 0.917 | 23.162 | 31.146 | 14.431 | 0.490 | 7.678 | 46.171 | 110.511 | 0.011 |
| SSO-PGA | **39.358** | **2.823** | **2.078** | **0.921** | **39.225** | **2.857** | **2.108** | **0.921** | **39.171** | **2.876** | **2.123** | **0.919** |

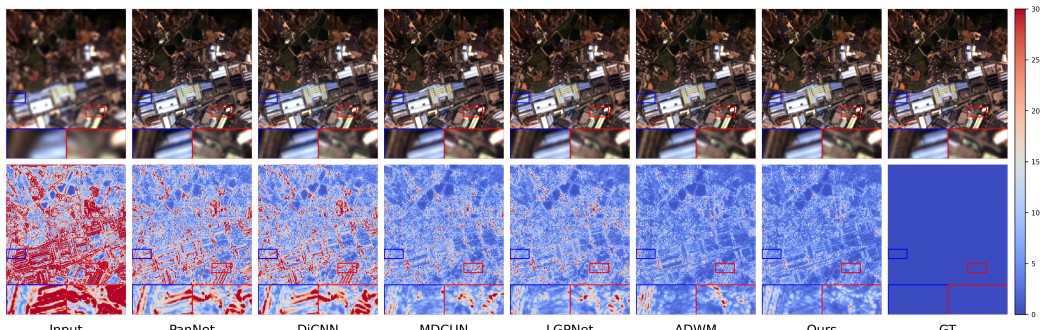

Figure 7: Visual comparison (the first row) and the corresponding error map (the second row) of our method and some representative methods on the GF2 reduced-resolution dataset.

## 4.2 COMPARISON WITH SOTAs

### 4.2.1 MULTISPECTRAL IMAGE FUSION

**Datasets and Setting.** We conducted experiments on three datasets consisting of satellite images captured by WorldView-3 (WV3), QuickBird (QB), and GaoFen-2 (GF2), provided by the PanCollection repository Deng et al. (2022). We evaluate our method using a set of widely used performance metrics. For reduced-resolution data, we use PSNR, SAM Boardman (1993), ERGAS Wald (2002), and Q4/Q8 Garzelli & Nencini (2009).

Table 3: Quantitative comparison for multispectral image fusion on reduced-resolution datasets: WV3, QB, and GF2. The best results are in **bold** and the second-best values are in underlined.

| Methods | WV3 | | | | QB | | | | GF2 | | | |
|---|---|---|---|---|---|---|---|---|---|---|---|---|
| | PSNR↑ | SAM↓ | ERGAS↓ | Q8↑ | PSNR↑ | SAM↓ | ERGAS↓ | Q4↑ | PSNR↑ | SAM↓ | ERGAS↓ | Q4↑ |
| MTF-GLP-FS Vivone et al. (2018) | 32.963 | 5.316 | 4.700 | 0.833 | 32.709 | 7.792 | 7.373 | 0.835 | 35.540 | 1.655 | 1.589 | 0.897 |
| BDSD-PC Vivone (2019) | 32.970 | 5.428 | 4.697 | 0.829 | 32.550 | 8.085 | 7.513 | 0.831 | 35.180 | 1.681 | 1.667 | 0.892 |
| TV Palsson et al. (2013) | 32.381 | 5.692 | 4.855 | 0.795 | 32.136 | 7.510 | 7.690 | 0.821 | 35.237 | 1.911 | 1.737 | 0.907 |
| PNN Masi et al. (2016) | 37.313 | 3.677 | 2.681 | 0.893 | 36.942 | 5.181 | 4.468 | 0.918 | 39.071 | 1.048 | 1.057 | 0.960 |
| PanNet Yang et al. (2017) | 37.346 | 3.613 | 2.664 | 0.891 | 34.678 | 5.767 | 5.859 | 0.885 | 40.243 | 0.997 | 0.919 | 0.967 |
| DiCNN He et al. (2019) | 37.390 | 3.592 | 2.672 | 0.900 | 35.781 | 5.367 | 5.133 | 0.904 | 38.906 | 1.053 | 1.081 | 0.959 |
| FusionNet Deng et al. (2020) | 38.047 | 3.324 | 2.465 | 0.904 | 37.540 | 4.904 | 4.156 | 0.925 | 39.639 | 0.974 | 0.988 | 0.964 |
| MDCUN Yang et al. (2022) | 37.973 | 3.298 | 2.479 | 0.909 | 36.178 | 4.963 | 4.698 | 0.915 | 41.138 | 0.870 | 0.815 | 0.974 |
| LAGNet Jin et al. (2022) | 38.592 | 3.103 | 2.291 | 0.910 | 38.209 | 4.534 | 3.812 | 0.934 | 42.735 | 0.786 | 0.687 | 0.980 |
| LGPNet Zhao et al. (2023) | 38.147 | 3.270 | 2.422 | 0.902 | 36.443 | 4.954 | 4.777 | 0.915 | 41.843 | 0.845 | 0.765 | 0.976 |
| U2Net Peng et al. (2023) | 39.117 | 2.888 | 2.149 | 0.920 | 38.065 | 4.642 | 3.987 | 0.931 | 43.379 | 0.714 | 0.632 | 0.981 |
| CANNet Duan et al. (2024) | 39.003 | 2.941 | 2.174 | 0.920 | 38.488 | 4.496 | 3.698 | 0.937 | 43.496 | 0.707 | 0.630 | 0.983 |
| PanMamba He et al. (2025) | 39.012 | 2.913 | 2.184 | 0.920 | 37.356 | 4.625 | 4.277 | 0.929 | 42.907 | 0.743 | 0.684 | 0.982 |
| ADWM Huang et al. (2025a) | 39.170 | 2.913 | 2.145 | **0.921** | 38.466 | 4.450 | 3.705 | 0.937 | 43.884 | 0.672 | 0.597 | **0.985** |
| SSO-PGA (ours) | **39.358** | **2.823** | **2.078** | **0.921** | **38.807** | **4.312** | **3.493** | **0.938** | **44.005** | **0.660** | **0.574** | **0.985** |

**Experimental Results.** As shown in Tab. 3, our proposed SSO-PGA consistently achieves the best results across all datasets compared to other methods. Specifically, in terms of PSNR, our method outperforms the second-best method by 0.188 dB, 0.319 dB, and 0.121 dB on the WV3, QB, and GF2 datasets, respectively. These consistent improvements validate the effectiveness of our deep

unfolding framework. Furthermore, Fig. 7 presents a qualitative visual comparison of the GF2 dataset against several representative methods. Our method produces reconstructions closer to the ground truth with lower residuals, further highlighting its superiority.

### 4.2.2 Flash Guided Non-Flash Image Denoising

**Datasets and Setting.** Following the experimental protocol in recent studies Deng et al. (2024); Xu et al. (2024), we used the following datasets for training and testing: the Flash and Ambient Illuminations Dataset (FAID) Aksoy et al. (2018) and the Multi-Illumination Dataset (MID) Murmann et al. (2019). We added varying levels of Gaussian noise to the non-flash images in each dataset and used PSNR as the evaluation metric.

**Experimental Results.** As shown in Tab. 4, our method outperforms the others on MID and FAID datasets. This not only highlights the performance of our method but also demonstrates its versatility and generalization capabilities across various tasks. It's worth noting that although our method's performance is on par with DeepM$^2$CDL Deng et al. (2024), our method has a parameter count of just 2.90M, which is significantly smaller than DeepM$^2$CDL Deng et al. (2024)'s 421.14M. This highlights the lightweight and efficient nature of our approach, as it minimizes computational overhead while maintaining comparable performance.

Table 4: Quantitative comparison for flash guided non-flash image denoising in terms of PSNR (dB) on MID and FAID datasets. The best results are in **bold** and the second-best values are underlined.

| Methods | | MID | | | FAID | | |
|---|---|---|---|---|---|---|---|
| | | $\sigma = 25$ | $\sigma = 50$ | $\sigma = 75$ | $\sigma = 25$ | $\sigma = 50$ | $\sigma = 75$ |
| DnCNN Zhang et al. (2017) | | 34.57 | 32.69 | 31.26 | 35.38 | 31.94 | 30.08 |
| DJFR Li et al. (2019a) | | 37.03 | 32.96 | 31.84 | 33.76 | 30.61 | 28.92 |
| CUNet Deng & Dragotti (2020) | | 34.61 | 32.39 | 31.18 | 35.86 | 33.05 | 31.30 |
| UMGF Shi et al. (2021) | | 38.18 | 35.84 | 34.30 | 34.52 | 31.81 | 30.43 |
| MN Xu et al. (2022) | | 39.51 | 37.01 | 35.50 | 36.15 | 33.34 | 31.83 |
| FGDNet Sheng et al. (2022) | | 38.38 | 35.88 | 34.39 | 34.99 | 32.15 | 30.81 |
| RIDFhF Oh et al. (2023) | | 38.31 | 35.33 | 33.74 | 36.25 | 33.48 | 31.92 |
| DeepM$^2$CDL Deng et al. (2024) | (Para: 421.14M) | 39.67 | 37.61 | **36.28** | 36.86 | **34.43** | **32.95** |
| SSO-PGA (ours) | (Para: 2.90M) | **39.84** | **37.66** | 35.71 | **36.88** | 34.12 | 32.92 |

### 4.3 Ablation Study

We conduct a comprehensive ablation study on the SSO-PGA network. First, we remove the $Prox_\phi(\cdot)$ module to construct the variant V-1. Then, we individually remove the standard convolution, dilated convolution, and frequency-domain convolution from the MSFB to construct variants V-2, V-3, and V-4, respectively. The results in Tab. 5 show that SSO-PGA outperforms variant V-1, which demonstrates the

Table 5: Ablation Study of different variants.

| Variant | PSNR↑ | SAM↓ | ERGAS↓ | Q8↑ |
|---|---|---|---|---|
| V-1 | 38.194 | 3.176 | 2.396 | 0.911 |
| V-2 | 39.301 | 2.849 | 2.088 | 0.920 |
| V-3 | 39.190 | 2.868 | 2.108 | 0.919 |
| V-4 | 39.236 | 2.854 | 2.106 | **0.921** |
| ours | **39.358** | **2.823** | **2.078** | **0.921** |

importance of the deep prior. Furthermore, the superiority of SSO-PGA over V-2, V-3, and V-4 verifies that each branch in the MSFB module is indispensable and plays a critical role in enabling comprehensive information fusion.

## 5 Conclusion

This paper proposes SSO-PGA, a novel multiplicative proximal gradient algorithm enhanced by Sliding Sigmoid Operator, which improves stability and adaptivity. We replace the traditional gradient descent step with a learnable sigmoid-based operator, which inherently enforces non-negativity and boundedness. SSO-PGA is formulated for multi-modal restoration. We then iteratively solve the model and further unfold it into a deep neural network. Both numerical and real-world experiments verify the superiority of SSO-PGA and its significant improvements in accuracy and convergence speed over conventional PGA. Future work will focus on analyzing the theoretical convergence rate of SSO-PGA and extending its application to broader vision tasks.

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

# A   APPENDIX

This supplementary material provides additional technical and experimental details that support the main paper. It is organized as follows:

- **Sec.A.1 Additional Proofs:** We provide detailed theoretical proofs of Theorem 1, Lemma 1, Lemma 2, Lemma 3, and Theorem 2 related to the SSO-PGA.
- **Sec.A.2 Limitations:** We discuss the known limitation of our work and how to address it.
- **Sec.A.3 Broader Impact:** We reflect on the potential applications and societal impact of our proposed method and framework.
- **Sec.A.4 Datasets:** We provide an overview of the datasets employed in this work.
- **Sec.A.5 Implementation Details:** We describe the compute resources, hyperparameters, and training strategies used in our experiments.
- **Sec.A.6 Experimental Results on Real-world Dataset:** We provide the experimental results on a full-resolution dataset to indicate the strong potential of our SSO-PGA for real-world applications.
- **Sec.A.7 Additional Comparison with Traditional Proximal Gradient Algorithm:** We provide additional comparison with traditional proximal gradient algorithm to validate the advantages of our method.
- **Sec.A.8 Additional Ablation Study:** We provide additional ablation studies to validate the effectiveness of each component of our method.
- **Sec.A.9 Additional Numerical Experiments:** We provide additional numerical experiment results to further validate the advantages of our method.
- **Sec.A.10 Additional Visual Experimental Results:** We include extended visual comparisons to further validate the effectiveness of our approach.
- **Sec.A.11 The Use of LLMs:** We describe the use of LLMs in our work.

## A.1   ADDITIONAL PROOFS

**Proof of Theorem 1**

*Proof.* Using the identity $\sigma(z) + \sigma(-z) = 1$, we have:

$$
\begin{aligned}
SSO_\alpha\left(\nabla\mathcal{E}(\boldsymbol{y}_i^{t-1})\right) - 1 &= 2\left[\sigma(-\nabla\mathcal{E}(\boldsymbol{y}_i^{t-1}) - \alpha) + \sigma(\alpha) - 1\right] \\
&= 2\left[\sigma(-\nabla\mathcal{E}(\boldsymbol{y}_i^{t-1}) - \alpha) - \sigma(-\alpha)\right].
\end{aligned}
\tag{25}
$$

According to the Lagrange Mean Value Theorem, there exists $\xi_i^t$ between $-\alpha$ and $-\nabla\mathcal{E}(\boldsymbol{y}_i^{t-1}) - \alpha$ such that

$$
\sigma(-\nabla\mathcal{E}(\boldsymbol{y}_i^{t-1}) - \alpha) - \sigma(-\alpha) = \left(-\nabla\mathcal{E}(\boldsymbol{y}_i^{t-1})\right)\sigma'(\xi_i^t),
\tag{26}
$$

where $\sigma'(z) = \sigma(z)[1 - \sigma(z)] \in (0, \frac{1}{4}], \quad \forall z \in \mathbb{R}$.

Therefore,

$$
SSO_\alpha\left(\nabla\mathcal{E}(\boldsymbol{y}_i^{t-1})\right) - 1 = -2\nabla\mathcal{E}(\boldsymbol{y}_i^{t-1})\,\sigma'(\xi_i^t).
\tag{27}
$$

Set $\theta_i^t = 2\sigma'(\xi_i^t)$. Then $SSO_\alpha\left(\nabla\mathcal{E}(\boldsymbol{y}_i^{t-1})\right) = 1 - \theta_i^t\nabla\mathcal{E}(\boldsymbol{y}_i^{t-1})$, and since $\sigma'(z) \in (0, \frac{1}{4}]$, it follows that $\theta_i^t \in (0, \frac{1}{2}]$. Thus, we have:

$$
\boldsymbol{y}_i^t = \boldsymbol{y}_i^{t-1} \cdot SSO_\alpha\left(\nabla\mathcal{E}(\boldsymbol{y}_i^{t-1})\right) = \boldsymbol{y}_i^{t-1} - \boldsymbol{y}_i^{t-1}\theta_i^t\nabla\mathcal{E}(\boldsymbol{y}_i^{t-1}).
\tag{28}
$$

Set $\rho_i = \boldsymbol{y}_i^{t-1}\theta_i^t$, proof complete. $\qquad\square$

**Proof of Lemma 1**

*Proof.* Recall that the sliding sigmoid operator is defined as:

$$SSO_\alpha(z) = 2\sigma(-z - \alpha) + 2\sigma(\alpha) - 1, \quad \text{where} \quad \sigma(u) = \frac{1}{1 + e^{-u}}. \tag{29}$$

Since $SSO_\alpha(0) = 1$, by the Lagrange Mean Value Theorem, for some $\xi \in (0, z)$ (or $(z, 0)$), we have:

$$SSO_\alpha(z) - 1 = SSO'_\alpha(\xi) \cdot z. \tag{30}$$

Now compute the derivative:

$$SSO'_\alpha(u) = \frac{d}{du}\left[2\sigma(-u - \alpha)\right] = -2\sigma(-u - \alpha)(1 - \sigma(-u - \alpha)). \tag{31}$$

The maximum of $\sigma(v)(1 - \sigma(v))$ over $v \in \mathbb{R}$ is $\frac{1}{4}$, hence:

$$|SSO'_\alpha(u)| \le \frac{1}{2} \le \frac{1 + \alpha}{2} = \eta(\alpha). \tag{32}$$

Thus:

$$|SSO_\alpha(z) - 1| \le |SSO'_\alpha(\xi)| \cdot |z| \le \eta(\alpha)|z|. \tag{33}$$
$\square$

**Proof of Lemma 2**

*Proof.* Consider the scalar function $\varphi(t) = \mathcal{E}(\boldsymbol{y} + t\boldsymbol{d}), \quad t \in [0, 1]$. We have:

$$\mathcal{E}(\boldsymbol{y} + \boldsymbol{d}) - \mathcal{E}(\boldsymbol{y}) = \varphi(1) - \varphi(0) = \int_0^1 \varphi'(t)\, dt = \int_0^1 \langle \nabla\mathcal{E}(\boldsymbol{y} + t\boldsymbol{d}), \boldsymbol{d} \rangle dt. \tag{34}$$

Add and subtract $\nabla\mathcal{E}(\boldsymbol{y})$ inside the inner product and apply Cauchy–Schwarz:

$$
\begin{aligned}
\mathcal{E}(\boldsymbol{y} + \boldsymbol{d}) - \mathcal{E}(\boldsymbol{y}) &= \int_0^1 \langle \nabla\mathcal{E}(\boldsymbol{y}), \boldsymbol{d} \rangle dt + \int_0^1 \langle \nabla\mathcal{E}(\boldsymbol{y} + t\boldsymbol{d}) - \nabla\mathcal{E}(\boldsymbol{y}), \boldsymbol{d} \rangle dt \\
&= \langle \nabla\mathcal{E}(\boldsymbol{y}), \boldsymbol{d} \rangle + \int_0^1 \langle \nabla\mathcal{E}(\boldsymbol{y} + t\boldsymbol{d}) - \nabla\mathcal{E}(\boldsymbol{y}), \boldsymbol{d} \rangle dt \\
&\le \langle \nabla\mathcal{E}(\boldsymbol{y}), \boldsymbol{d} \rangle + \int_0^1 \|\nabla\mathcal{E}(\boldsymbol{y} + t\boldsymbol{d}) - \nabla\mathcal{E}(\boldsymbol{y})\|_2 \|\boldsymbol{d}\|_2\, dt \\
&\le \langle \nabla\mathcal{E}(\boldsymbol{y}), \boldsymbol{d} \rangle + \int_0^1 L\, t\, \|\boldsymbol{d}\|_2^2\, dt \quad \text{(by } L\text{–Lipschitzness)} \\
&= \langle \nabla\mathcal{E}(\boldsymbol{y}), \boldsymbol{d} \rangle + \frac{L}{2} \|\boldsymbol{d}\|_2^2.
\end{aligned}
\tag{35}
$$

Thus, proof complete. $\square$

**Proof of Lemma 3**

*Proof.* The gradient of the objective is:

$$\nabla\mathcal{E}(\boldsymbol{y}) = 2\boldsymbol{H}^\top(\boldsymbol{H}\boldsymbol{y} - \boldsymbol{x}). \tag{36}$$

So for any $\boldsymbol{y}, \boldsymbol{z}$, we have:

$$
\begin{aligned}
\|\nabla\mathcal{E}(\boldsymbol{y}) - \nabla\mathcal{E}(\boldsymbol{z})\|_2 &= 2\|\boldsymbol{H}^\top\boldsymbol{H}(\boldsymbol{y} - \boldsymbol{z})\|_2 \\
&\le 2\|\boldsymbol{H}^\top\boldsymbol{H}\|_2 \cdot \|\boldsymbol{y} - \boldsymbol{z}\|_2 \\
&= 2\|\boldsymbol{H}\|_2^2 \cdot \|\boldsymbol{y} - \boldsymbol{z}\|_2.
\end{aligned}
\tag{37}
$$

Thus,

$$\|\nabla\mathcal{E}(\boldsymbol{y}) - \nabla\mathcal{E}(\boldsymbol{z})\|_2 \le L\|\boldsymbol{y} - \boldsymbol{z}\|_2, \quad L = 2\|\boldsymbol{H}\|_2^2. \tag{38}$$
$\square$

**Proof of Theorem 2**

*Proof.* Fix $t$, and denote $\boldsymbol{y} = \boldsymbol{y}^{t-1}$, $\boldsymbol{y}^+ = \boldsymbol{y}^t$, $\boldsymbol{g} = \nabla\mathcal{E}(\boldsymbol{y})$ and $\boldsymbol{s} = SSO_\alpha(\boldsymbol{g}) - \boldsymbol{1}$ for simplicity.. From Eq. (14), we have $\boldsymbol{y}^+ = \boldsymbol{y} + \boldsymbol{d}$ with $\boldsymbol{d} = \boldsymbol{y} \odot \boldsymbol{s}$. Then, we have:

$$\langle \boldsymbol{g}, \boldsymbol{d} \rangle = -\sum_i |d_i||g_i|, \tag{39}$$

**Lemma 1** with $z = g_i$ yields $|s_i| \le \eta(\alpha)|g_i|$. Hence:

$$\|\boldsymbol{d}\|_2^2 = \sum_i |d_i||s_i|y_i \le \eta(\alpha)\sum_i |d_i||g_i|y_i. \tag{40}$$

From **Lemma 3**, $\alpha \le 2/(\kappa\|\boldsymbol{H}\|_2^2) - 1 = 4/(\kappa L) - 1$, we have $\eta(\alpha) = (\alpha + 1)/2 \le 2/(\kappa L)$. Combining this with the bound on $\|\boldsymbol{d}\|_2^2$ gives:

$$\frac{L}{2}\|\boldsymbol{d}\|_2^2 \le \sum_i |d_i||g_i| = -\langle \boldsymbol{g}, \boldsymbol{d} \rangle. \tag{41}$$

Inserting the bounds into **Lemma 2**:

$$\mathcal{E}(\boldsymbol{y}^+) - \mathcal{E}(\boldsymbol{y}) \le \langle \boldsymbol{g}, \boldsymbol{d} \rangle + \frac{L}{2}\|\boldsymbol{d}\|_2^2 \tag{42}$$

$$\le \langle \boldsymbol{g}, \boldsymbol{d} \rangle - \langle \boldsymbol{g}, \boldsymbol{d} \rangle = 0. \tag{43}$$

Thus, $\mathcal{E}(\boldsymbol{y}^t) \le \mathcal{E}(\boldsymbol{y}^{t-1})$ for every $t \ge 1$. □

### A.2 LIMITATIONS

A limitation of our study is that SSO-PGA performs well when the solution to the optimization problem lies between 0 and 1, but exhibits oscillatory, non-convergent behavior when the true solution is large. For example, when we set the optimal solution to 6, as shown in Fig. 8 and Fig. 9, this issue becomes apparent.

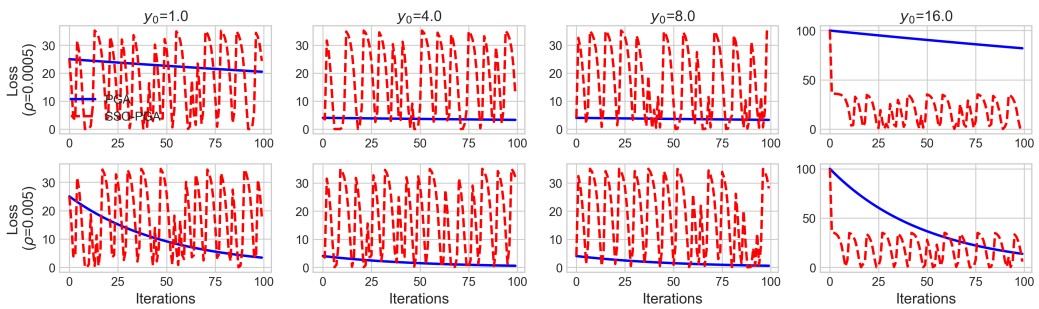

Figure 8: Comparison of numerical simulation results for SSO-PGA and PGA on Problem I when the true solution is large.

This specific instability is circumvented when SSO-PGA is integrated with a deep network. This is because, in deep learning, it's standard practice to normalize network inputs and outputs to the [0, 1] range. The final results are then obtained through inverse normalization. This preprocessing step naturally prevents the instability observed with large solution values.

Furthermore, we've identified that this oscillatory behavior is caused by excessively large gradients. We propose a straightforward solution to mitigate this problem during the optimization process: gradient clipping. For instance, by clipping the gradients of SSO-PGA to a range of [-0.1, 0.1], as shown in Fig. 10 and Fig. 11, SSO-PGA still demonstrates a faster convergence rate compared to PGA.

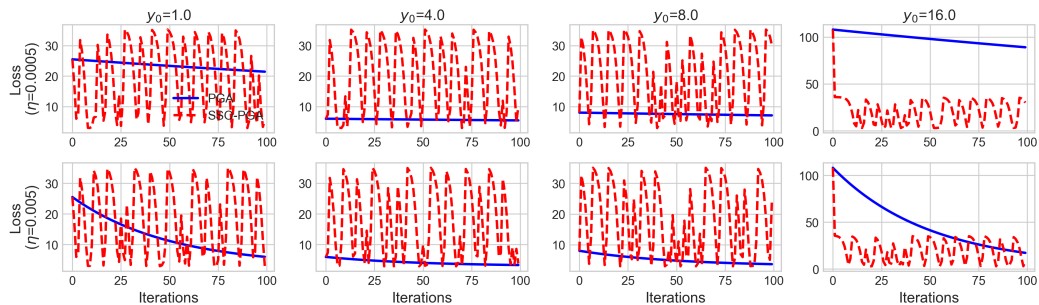

Figure 9: Comparison of numerical simulation results for SSO-PGA and PGA on Problem II when the true solution is large.

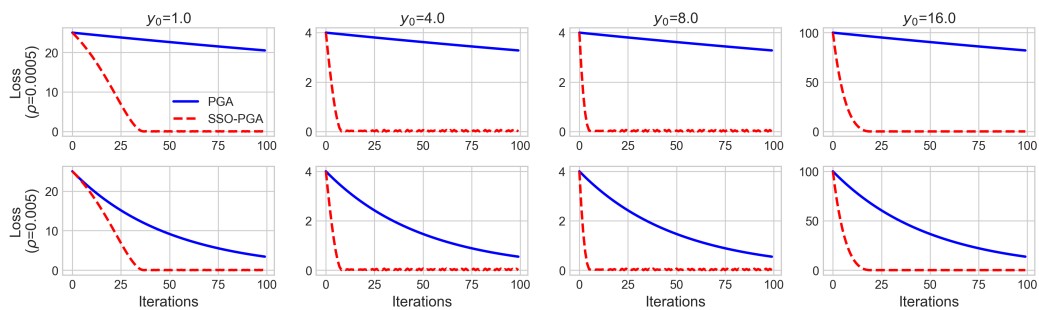

Figure 10: Comparison of numerical simulation results for SSO-PGA (with gradient clipping) and PGA on Problem I when the true solution is large.

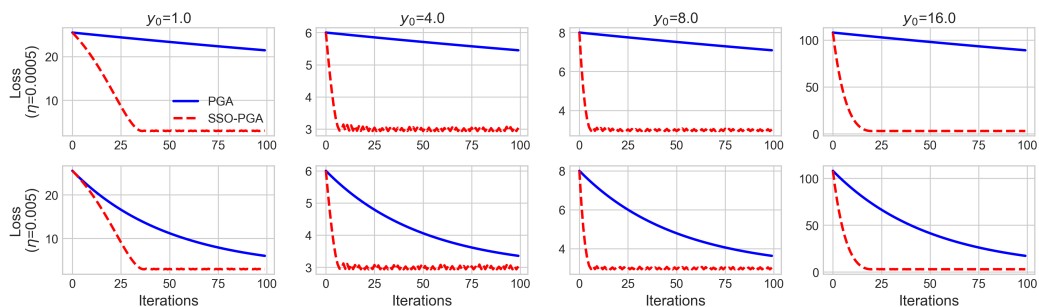

Figure 11: Comparison of numerical simulation results for SSO-PGA (with gradient clipping) and PGA on Problem II when the true solution is large.

## A.3 BROADER IMPACT

The proposed SSO-PGA framework offers a robust and interpretable optimization strategy that extends beyond the task of multispectral image fusion and flash guided non-flash image denoising. The SSO-PGA framework is built upon a gradient-based update mechanism that naturally enforces non-negativity, making it easily adaptable to various inverse problems in computer vision and image reconstruction tasks. These include, but are not limited to, image deblurring, denoising, super-resolution, compressive sensing reconstruction, and medical image enhancement. The deep unfolding nature of SSO-PGA not only enables convergence-guaranteed iterative learning but also offers structural transparency, which is particularly desirable in safety-critical applications like healthcare and autonomous navigation. The strong empirical performance and theoretical convergence guarantee of SSO-PGA make it a promising foundation for future research on interpretable and robust optimization in deep learning systems.

## A.4 DATASETS

In our experiments on multispectral image fusion, we utilized remote sensing image datasets from the PanCollection repository Deng et al. (2022), encompassing three satellite sources: WorldView-3 (WV3), QuickBird (QB), and GaoFen-2 (GF2). Each dataset is divided into training and testing subsets. A detailed summary of the sample counts and image dimensions under both reduced- and full-resolution settings is provided in Table 6.

Table 6: Summary of WorldView-3 (WV3), QuickBird (QB), and GaoFen-2 (GF2) datasets.

| Dataset | Samples | Image Size (PAN / LRMS / GT) |
|---------|---------|------------------------------|
| *Reduced-Resolution* | | |
| WV3 | 10,000 (train) / 20 (test) | $64{\times}64$ / $16{\times}16{\times}8$ / $64{\times}64{\times}8$ |
| QB | 17,000 (train) / 20 (test) | $64{\times}64$ / $16{\times}16{\times}4$ / $64{\times}64{\times}4$ |
| GF2 | 20,000 (train) / 20 (test) | $64{\times}64$ / $16{\times}16{\times}4$ / $64{\times}64{\times}4$ |
| *Full-Resolution* | | |
| WV3 | 20 (test) | $512{\times}512$ / $128{\times}128{\times}8$ / None |
| QB | 20 (test) | $512{\times}512$ / $128{\times}128{\times}4$ / None |
| GF2 | 20 (test) | $512{\times}512$ / $128{\times}128{\times}4$ / None |

In our experiments on flash guided non-flash image denoising, we utilized two common datasets: the Flash and Ambient Illuminations Dataset (FAID) Aksoy et al. (2018) and the Multi-Illumination Dataset (MID) Murmann et al. (2019). Each dataset is divided into training and testing subsets. A detailed summary of the sample counts and image dimensions is provided in Table 7.

Table 7: Summary of the Flash and Ambient Illuminations Dataset (FAID) Aksoy et al. (2018) and the Multi-Illumination Dataset (MID) Murmann et al. (2019).

| Dataset | Samples | Image Size |
|---------|---------|------------|
| FAID | 404 (train) / 12 (test) | $900{\times}600 \times 3$ |
| MID | 983 (train) / 30 (test) | $1500{\times}1000 \times 3$ |

## A.5 IMPLEMENTATION DETAILS

All training procedures are conducted on a high-performance computing server equipped with 8 NVIDIA RTX 4090 GPUs. Our training pipeline is implemented in Python 3.8.20 with PyTorch 2.4.1 + cu121, leveraging CUDA 12.1 for efficient GPU acceleration.

For multispectral image fusion, we employ the Adam optimizer Kingma & Ba (2014) with an initial learning rate of $1 \times 10^{-3}$ and a weight decay of $1 \times 10^{-8}$, and the learning rate is halved every 100 epochs. The model is trained for 300 epochs. During training, we apply dropout regularization with rates of 0.1 on the WV3 and QB datasets, and 0.25 on the GF2 dataset. To ensure high-quality reconstruction, we adopt a batch size of 32 throughout the training process. The entire model contains approximately 1.07 million trainable parameters and requires around 15.20 GiB of GPU memory. We compare our method with several state-of-the-art methods, including 3 traditional algorithms: MTF-GLP-FS Vivone et al. (2018), BDSD-PC Vivone (2019), and TV Palsson et al. (2013), and 11 deep learning/unfolding-based models: PNN Masi et al. (2016), PanNet Yang et al. (2017), DiCNN He et al. (2019), FusionNet Deng et al. (2020), MDCUN Yang et al. (2022), LAGNet Jin et al. (2022), LGPNet Zhao et al. (2023), U2Net Peng et al. (2023), CANNet Duan et al. (2024), PanMamba He et al. (2025), and ADWM Huang et al. (2025a).

For flash guided non-flash image denoising, we employ the Adam optimizer Kingma & Ba (2014) with an initial learning rate of $1 \times 10^{-3}$ and a weight decay of $1 \times 10^{-8}$, and the learning rate is

halved every 300 epochs. The model is trained for 2000 epochs. To ensure high-quality reconstruction, we adopt a batch size of 16 and a patch size of $128 \times 128$ throughout the training process. The entire model contains approximately 2.90 million trainable parameters and requires around 39.91 GiB of GPU memory. We compared our results against the following representative methods: DnCNN Zhang et al. (2017), DJFR Li et al. (2019a), CUNet Deng & Dragotti (2020), UMGF Shi et al. (2021), MN Xu et al. (2022), FGDNet Sheng et al. (2022), RIDFhF Oh et al. (2023), and DeepM$^2$CDL Deng et al. (2024).

### A.6 EXPERIMENTAL RESULTS ON REAL-WORLD DATASET

Following Huang et al. (2025b), for full-resolution data, we apply $D_s$, $D_\lambda$, and HQNR Arienzo et al. (2022) as the evaluation metric, which collectively provide a comprehensive measure of image fusion quality. We evaluate SSO-PGA on the full-resolution WV3 dataset, where it demonstrates significant advantages in Tab. 8. This outstanding performance not only validates the effectiveness of our method but also underscores its robustness and profound potential for real-world applications requiring high-fidelity image fusion.

Table 8: Quantitative comparison on WV3 dataset with 20 full-resolution samples. The best results are in **bold** and the second-best values are underlined.

| Methods | BDSD-PC Vivone (2019) | TV Palsson et al. (2013) | PNN Masi et al. (2016) | PanNet Yang et al. (2017) |
|---|---|---|---|---|
| $D_\lambda \downarrow$ | 0.063 | 0.023 | 0.021 | **0.017** |
| $D_s \downarrow$ | 0.073 | 0.039 | 0.043 | 0.047 |
| HQNR $\uparrow$ | 0.870 | 0.938 | 0.937 | 0.937 |

| Methods | DiCNN He et al. (2019) | LAGNet Jin et al. (2022) | LGPNet Zhao et al. (2023) | U2Net Peng et al. (2023) |
|---|---|---|---|---|
| $D_\lambda \downarrow$ | 0.036 | 0.037 | 0.022 | 0.020 |
| $D_s \downarrow$ | 0.046 | 0.042 | 0.039 | 0.028 |
| HQNR $\uparrow$ | 0.920 | 0.923 | 0.940 | 0.952 |

| Methods | CANNet Duan et al. (2024) | PanMamba He et al. (2025) | ADWM Huang et al. (2025a) | SSO-PGA (ours) |
|---|---|---|---|---|
| $D_\lambda \downarrow$ | 0.020 | 0.018 | 0.024 | 0.022 |
| $D_s \downarrow$ | 0.030 | 0.053 | 0.029 | **0.026** |
| HQNR $\uparrow$ | 0.951 | 0.930 | 0.948 | **0.953** |

### A.7 ADDITIONAL COMPARISON WITH TRADITIONAL PROXIMAL GRADIENT ALGORITHM

In this subsection, we provide a supplementary perturbation analysis for both PGA and SSO-PGA. Additionally, we present further experimental results for SSO-PGA at different iteration counts, as detailed in Tab. 9.

**Perturbation Analysis.** Fig. 12 and Tab. 10 present the comparison between SSO-PGA and PGA under varying levels of missing MS input (10%, 20%, and 50%) on the WV3 dataset. Across all perturbation levels, SSO-PGA consistently yields superior visual reconstruction and achieves higher PSNR and Q8 scores compared to PGA. Especially under a high missing rate (50%), the Q8 value of PGA drops to only 0.901, while SSO-PGA still maintains a result of 0.910. This demonstrates the strong robustness of the proposed SSO-PGA method in handling degraded and incomplete inputs.

In conclusion, comparing SSO-PGA with the PGA baseline, the results in Tab. 9 validate that SSO-PGA achieves faster and more stable convergence, while the perturbation experiments in Tab. 10 confirm its robustness under various missing ratios.

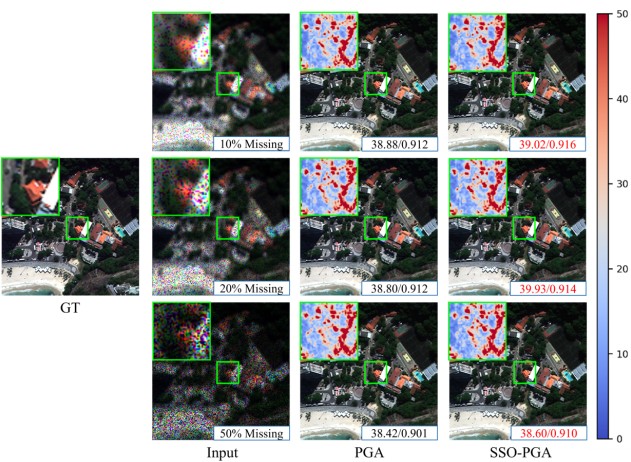

Figure 12: Visual comparison along with the corresponding PSNR and Q8 values of SSO-PGA and PGA on the WV3 dataset under varying missing ratios.

Table 9: Quantitative comparison of SSO-PGA and PGA on the WV3 reduced-resolution dataset over different iterations. The better results are in **bold**.

|  | PGA | | | | SSO-PGA | | | |
|---|---|---|---|---|---|---|---|---|
| Iteration 1 | PSNR↑ | SAM↓ | ERGAS↓ | Q8↑ | PSNR↑ | SAM↓ | ERGAS↓ | Q8↑ |
|  | 38.882 | 2.961 | **2.193** | 0.916 | **38.900** | **2.960** | 2.200 | **0.917** |
| Iteration 2 | PSNR↑ | SAM↓ | ERGAS↓ | Q8↑ | PSNR↑ | SAM↓ | ERGAS↓ | Q8↑ |
|  | 39.027 | 2.944 | 2.160 | 0.916 | **39.131** | **2.892** | **2.138** | **0.918** |
| Iteration 3 | PSNR↑ | SAM↓ | ERGAS↓ | Q8↑ | PSNR↑ | SAM↓ | ERGAS↓ | Q8↑ |
|  | 39.114 | 2.913 | 2.142 | 0.919 | **39.256** | **2.855** | **2.104** | **0.920** |
| Iteration 4 | PSNR↑ | SAM↓ | ERGAS↓ | Q8↑ | PSNR↑ | SAM↓ | ERGAS↓ | Q8↑ |
|  | 39.145 | 2.925 | 2.129 | 0.918 | **39.358** | **2.823** | **2.078** | **0.921** |
| Iteration 5 | PSNR↑ | SAM↓ | ERGAS↓ | Q8↑ | PSNR↑ | SAM↓ | ERGAS↓ | Q8↑ |
|  | 39.125 | 2.916 | 2.139 | 0.918 | **39.374** | **2.818** | **2.072** | **0.921** |

Table 10: Quantitative comparison of SSO-PGA and PGA on the WV3 reduced-resolution dataset under varying missing ratios. The better results are in **bold**.

|  | PGA | | | | SSO-PGA | | | |
|---|---|---|---|---|---|---|---|---|
| Missing 10% | PSNR↑ | SAM↓ | ERGAS↓ | Q8↑ | PSNR↑ | SAM↓ | ERGAS↓ | Q8↑ |
|  | 38.875 | 3.039 | 2.196 | 0.912 | **39.016** | **2.931** | **2.157** | **0.916** |
| Missing 20% | PSNR↑ | SAM↓ | ERGAS↓ | Q8↑ | PSNR↑ | SAM↓ | ERGAS↓ | Q8↑ |
|  | 38.804 | 3.056 | 2.210 | 0.912 | **38.928** | **2.972** | **2.184** | **0.914** |
| Missing 30% | PSNR↑ | SAM↓ | ERGAS↓ | Q8↑ | PSNR↑ | SAM↓ | ERGAS↓ | Q8↑ |
|  | 38.420 | 3.415 | 2.301 | 0.901 | **38.599** | **3.079** | **2.270** | **0.910** |

## A.8 ADDITIONAL ABLATION STUDY

**Different Sliding Parameter Settings.** Besides the parameter learning method described in the deep network architecture, there are two other ways to set the sliding parameter: manually fixed value and automated learning via a simple neural network (with a Convolution layer, a Sigmoid activation, another Convolution layer, and finally a Softplus activation). We've conducted additional experiments to compare these two approaches (Tab. 11), where the SSO-PGA-1 and SSO-PGA-0.1 are our method with fixed $\alpha$ values (1/0.1), and SSO-PGA-Auto is the automated way. From the table, we can observe that the performance of the fixed-value sliding parameter and the automated approach is slightly lower than that of our method in the paper.

Table 11: Comparison of Different Sliding Parameter Settings.

|              | PSNR ↑ | SAM ↓ | ERGAS ↓ | Q2N ↓ |
|--------------|--------|-------|---------|-------|
| SSO-PGA-Auto | 39.280 | 2.841 | 2.099   | **0.921** |
| SSO-PGA-1    | 39.287 | 2.824 | 2.092   | **0.921** |
| SSO-PGA-0.1  | 39.147 | 2.884 | 2.115   | 0.920 |
| SSO-PGA      | **39.358** | **2.823** | **2.078** | **0.921** |

**Comparison with Traditional Projected Operator.** To compare with traditional post-projection methods, we attempted to enforce non-negativity by applying activation functions (ReLU and Softplus) as projection operations after the gradient descent step in traditional PGA. The experimental results are shown in Tab. 12. However, both approaches performed even worse than PGA. The reason for this is that while these projection methods enforce non-negativity, they unfortunately lose information from negative values and alter the original gradient information during the process. In contrast, SSO-PGA guarantees non-negativity through a direct mapping while fully preserving the gradient information.

Table 12: Comparison with Traditional Projected Gradient Descent Methods.

|              | PSNR ↑ | SAM ↓ | ERGAS ↓ | Q2N ↓ |
|--------------|--------|-------|---------|-------|
| ReLU-PGA     | 36.600 | 3.557 | 2.825   | 0.900 |
| Softplus-PGA | 38.957 | 2.926 | 2.167   | 0.916 |
| PGA          | 39.145 | 2.925 | 2.129   | 0.918 |
| SSO-PGA      | **39.358** | **2.823** | **2.078** | **0.921** |

## A.9 ADDITIONAL NUMERICAL EXPERIMENTS

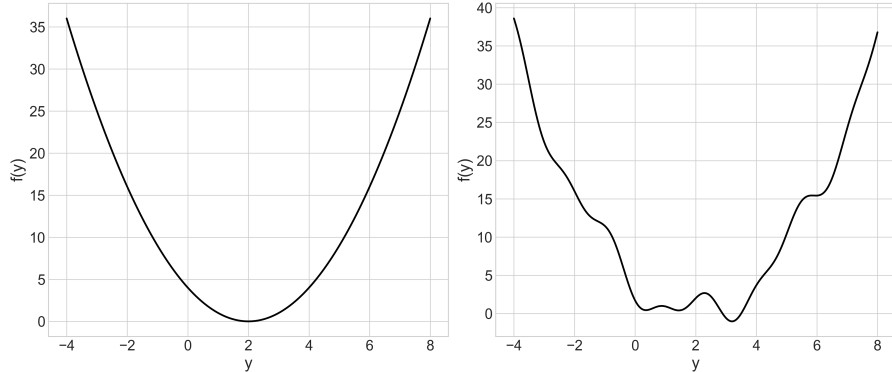

Figure 13: Landscapes for Problem I (left) and Problem I+ (right).

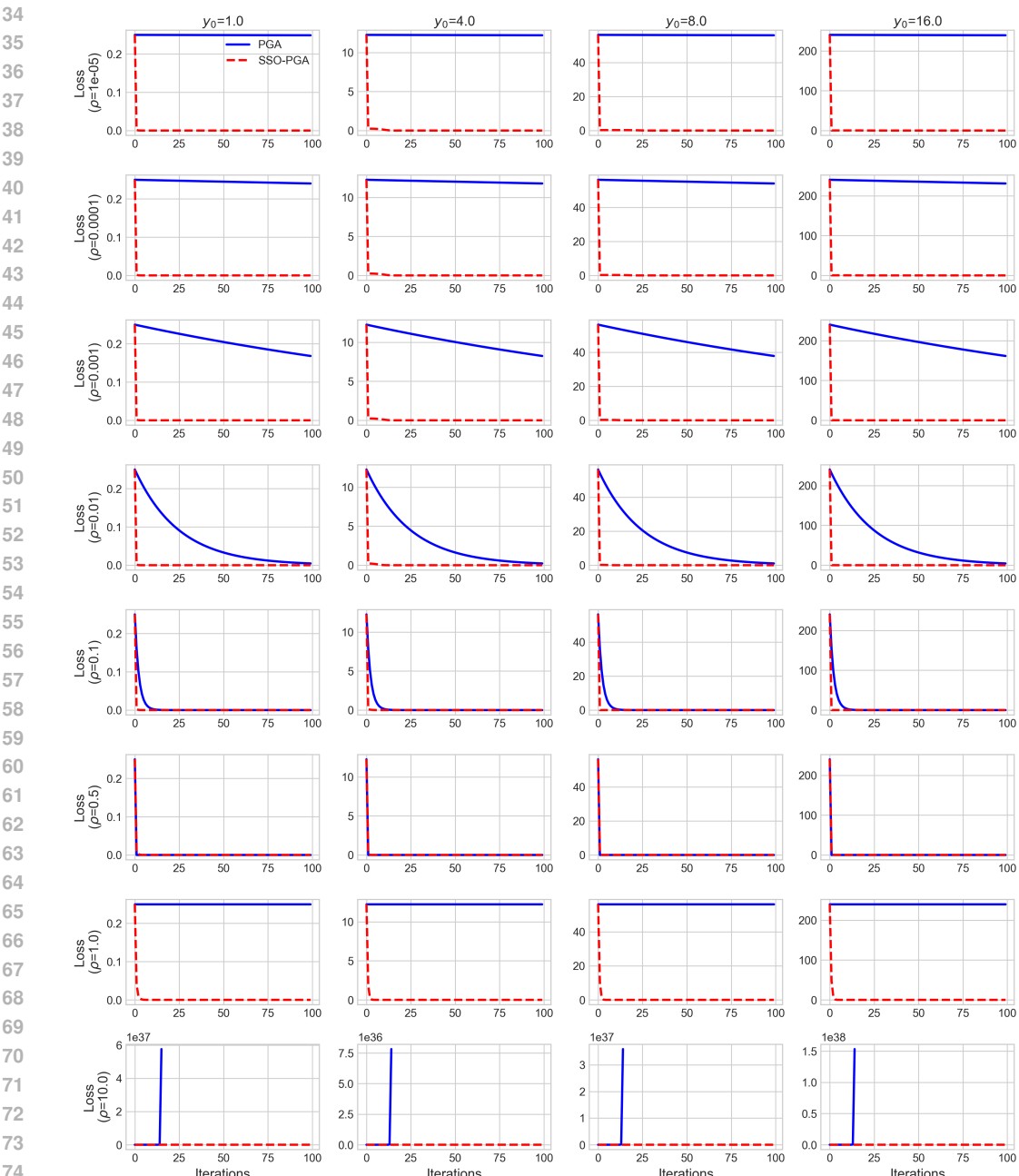

Figure 14: Additional comparison of numerical simulation results for SSO-PGA and PGA on Problem I.

In this subsection, we provide additional numerical simulation experiments. Specifically, in addition to the two problems from Eq. (24), we include two non-convex problems, denoted as Problem I+ and Problem II+:

$$\min_y (y - 0.5)^2 + \sin(4(x - 0.5)) + \cos(2(x - 0.5)), \qquad \text{(Problem I+)},$$

$$\min_y (y - 0.5)^2 + \sin(4(x - 0.5)) + \cos(2(x - 0.5)) + \frac{1}{2}|y|, \qquad \text{(Problem II+)}.$$

(44)

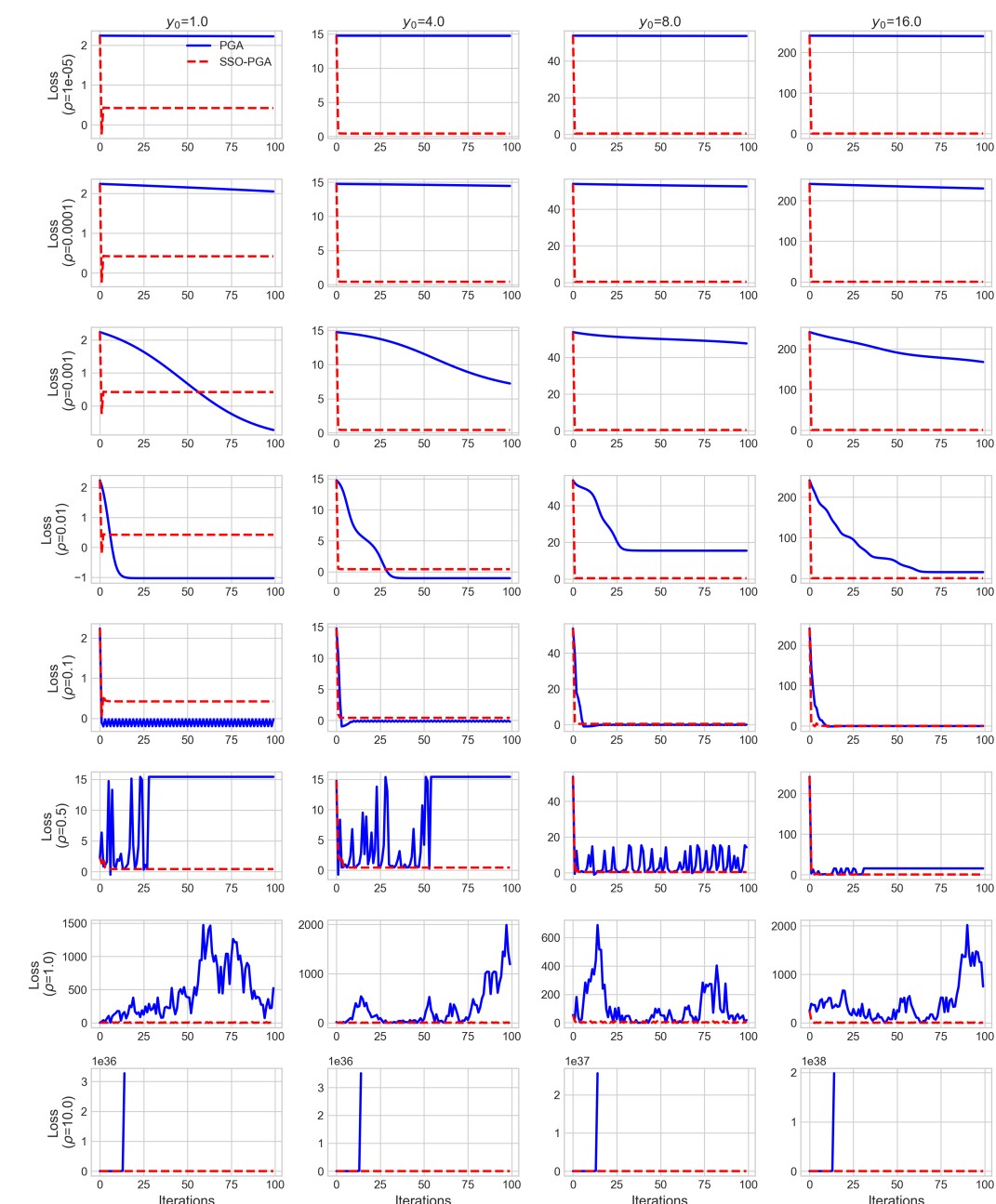

Figure 15: Additional comparison of numerical simulation results for SSO-PGA and PGA on Problem I+.

Fig. 13 shows the landscapes for Problem I (left) and Problem I+ (right), respectively. We tested a wide range of learning rates: 1e-5, 1e-4, 1e-3, 1e-2, 1e-1, 5e-1, 1, and 10. As shown in Fig. 14 to Fig. 17, our SSO-PGA consistently outperforms the traditional PGA under most parameter settings. This holds true for both convex and non-convex problems (Problem I and II, and their non-convex counterparts). We can observe that SSO-PGA is less sensitive to the learning rate. When the learning rate is small, SSO-PGA converges much faster than PGA. When the learning rate is large, SSO-PGA is more stable than PGA, especially with very large learning rates where PGA fails to converge. Additionally, in non-convex scenarios, SSO-PGA shows a slight advantage in avoiding local minima.

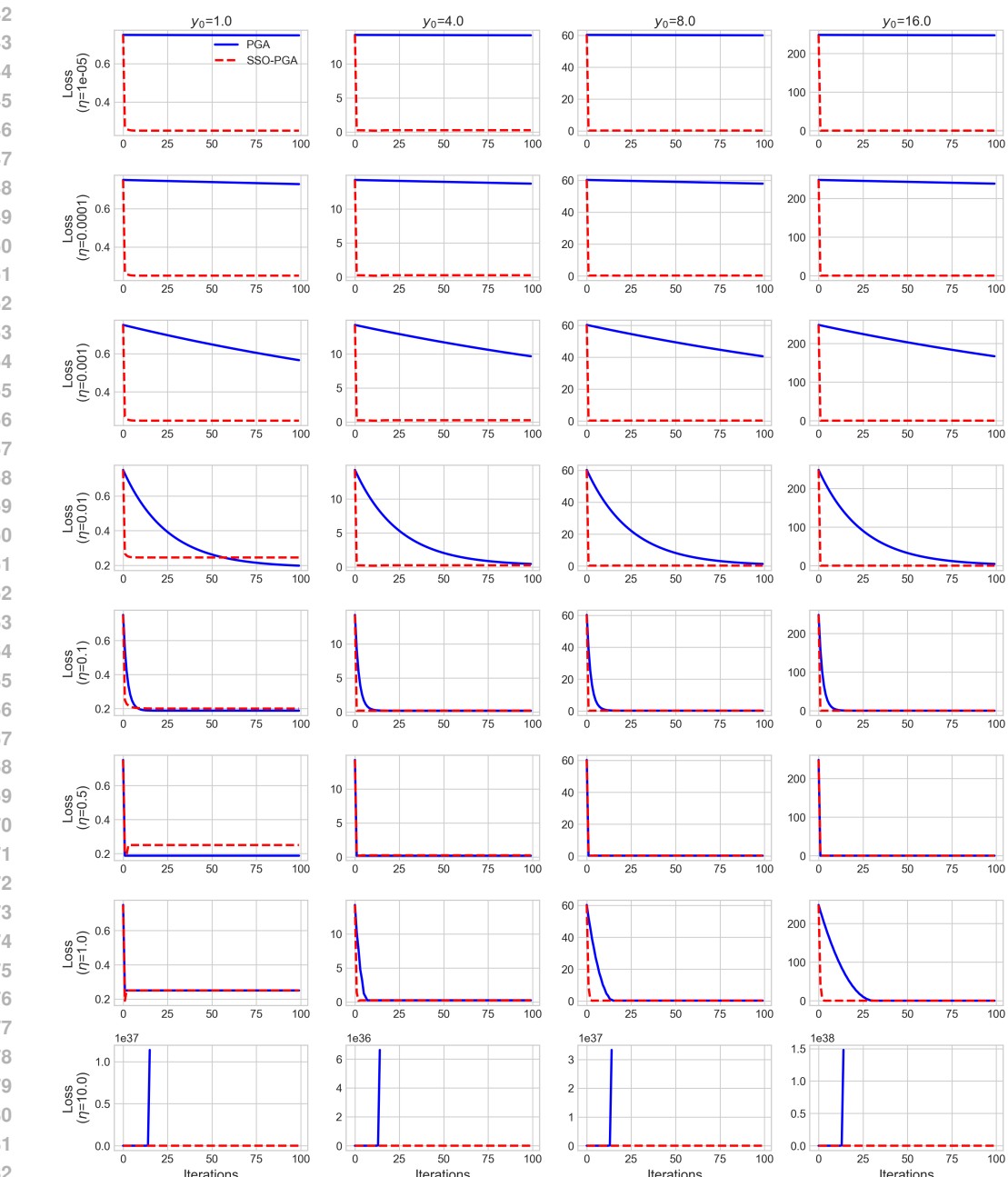

Figure 16: Additional comparison of numerical simulation results for SSO-PGA and PGA on Problem II.

This success is a direct result of the inherent advantages of the multiplicative update rule introduced by our novel SSO operator. By replacing the traditional subtractive gradient descent step with a sigmoid-based multiplicative update, our algorithm fundamentally transforms the optimization process, making it more stable, less sensitive to hyperparameters, and capable of achieving superior results. It's important to note that since this paper focuses on non-negative inverse problems, the optimal solutions in our numerical simulations are all greater than zero. If the optimal solution were less than zero, it would fall outside the scope of our study, and SSO-PGA would not be able to solve it.

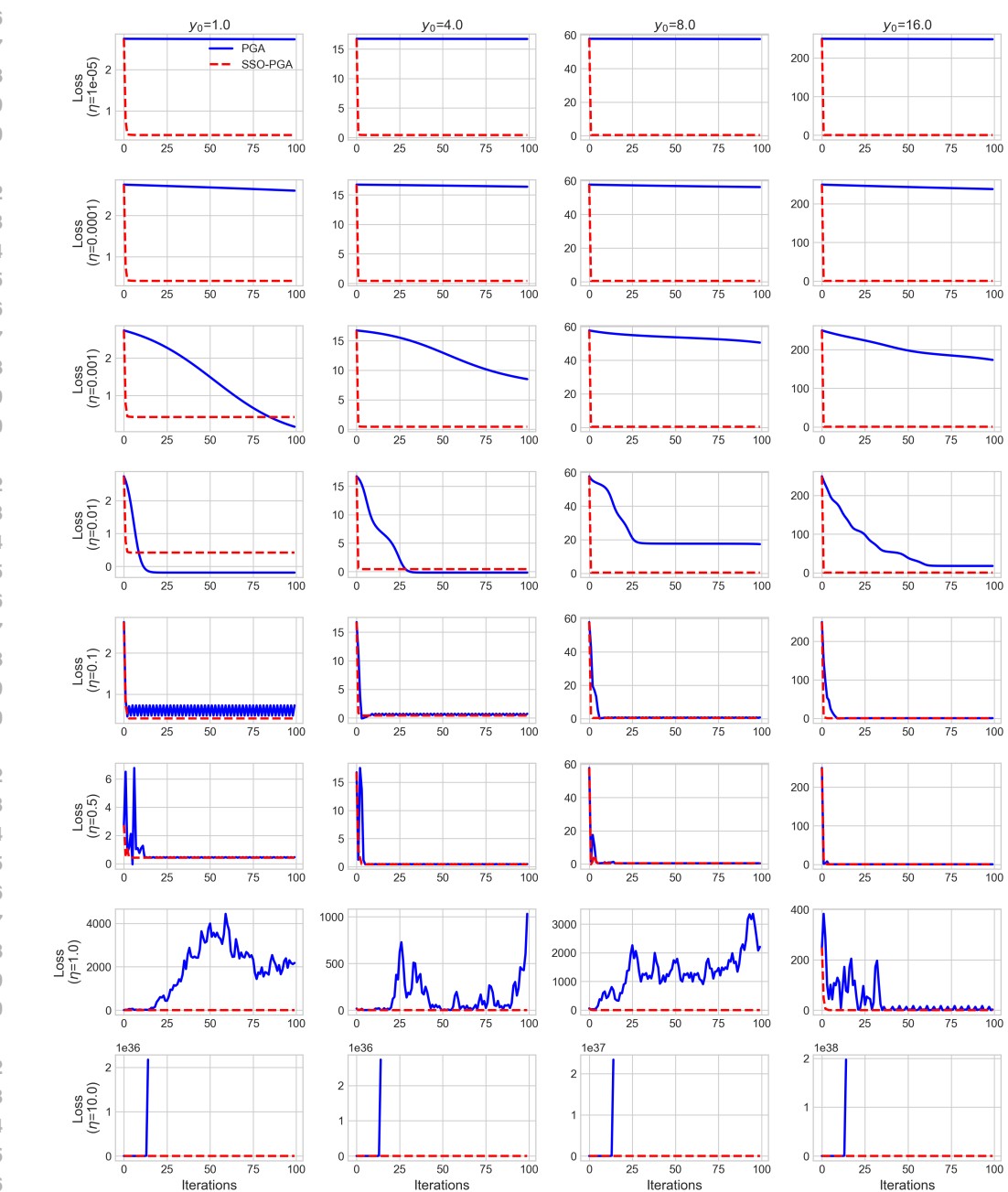

Figure 17: Additional comparison of numerical simulation results for SSO-PGA and PGA on Problem II+.

## A.10 ADDITIONAL VISUAL EXPERIMENTAL RESULTS

In this subsection, we present additional experimental results to further demonstrate the effectiveness and robustness of our proposed SSO-PGA method. The results cover the following aspects:

- **Qualitative Comparison on Flash Guided Non-Flash Image Denoising (Fig. 18, and Fig. 19:** Visual comparisons between SSO-PGA and several representative SOTA methods are provided across the two benchmark datasets (FAID and MID). These results clearly demonstrate that SSO-PGA consistently achieves superior denoising performance compared to other methods, yielding results that are closer to the ground truth.

- **Qualitative Comparison on Multispectral Image Fusion (Fig. 20, Fig. 21, and Fig. 22):** Visual comparisons between SSO-PGA and several representative SOTA methods are provided across the three benchmark datasets (WV3, QB, and GF2). These results clearly show that SSO-PGA consistently reconstructs sharper spatial details and produces reconstructions closer to the ground truth with lower residual.

- **Visualization Under Different Iteration Steps (Fig. 23 and Fig. 24):** We further present the reconstructed outputs of both SSO-PGA and the PGA baseline under varying numbers of iterations. The results demonstrate that SSO-PGA achieves high-fidelity fusion even with fewer unfolding steps and maintains performance when increasing the number of iterations, unlike the PGA baseline, which may suffer from degradation.

- **SSO vs. Gradient Descent Visualization (Fig. 25, Fig. 26, and Fig. 27):** We provide side-by-side visual comparisons of SSO-based and gradient-descent-based models, namely SSO-PGA vs. PGA baseline, and SSO-MDCUN vs. MDCUN Yang et al. (2022), across all datasets. The SSO-enhanced variants consistently produce better reconstruction with fewer spectral distortions and residual artifacts.

These extended experimental results collectively confirm the superiority of our proposed SSO-PGA framework in terms of reconstruction accuracy, convergence stability, and robustness across different scenarios.

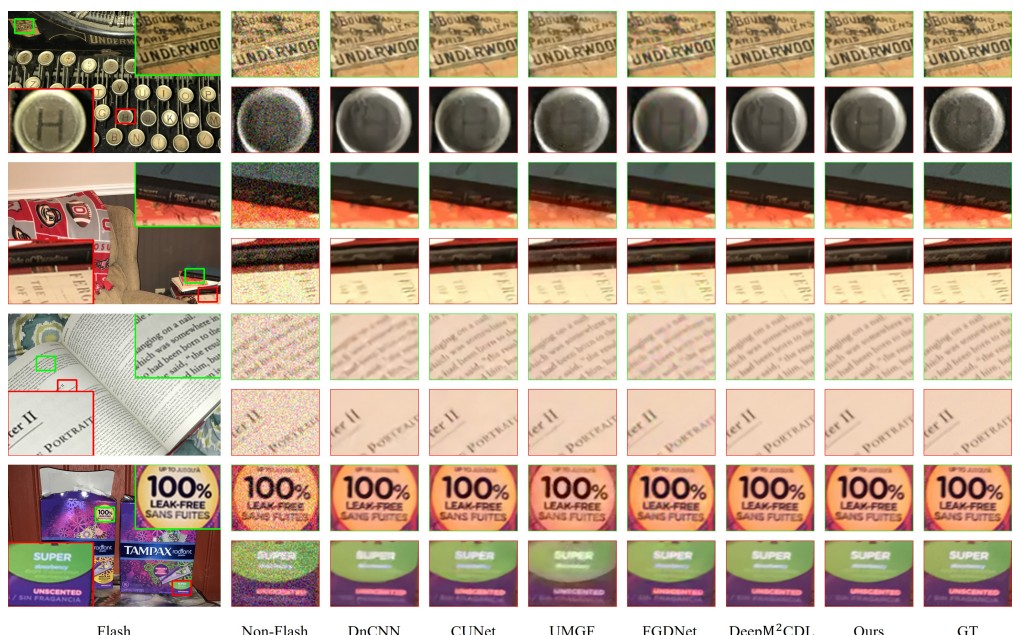

Figure 18: Visual comparison of our method and some representative methods on the FAID dataset.

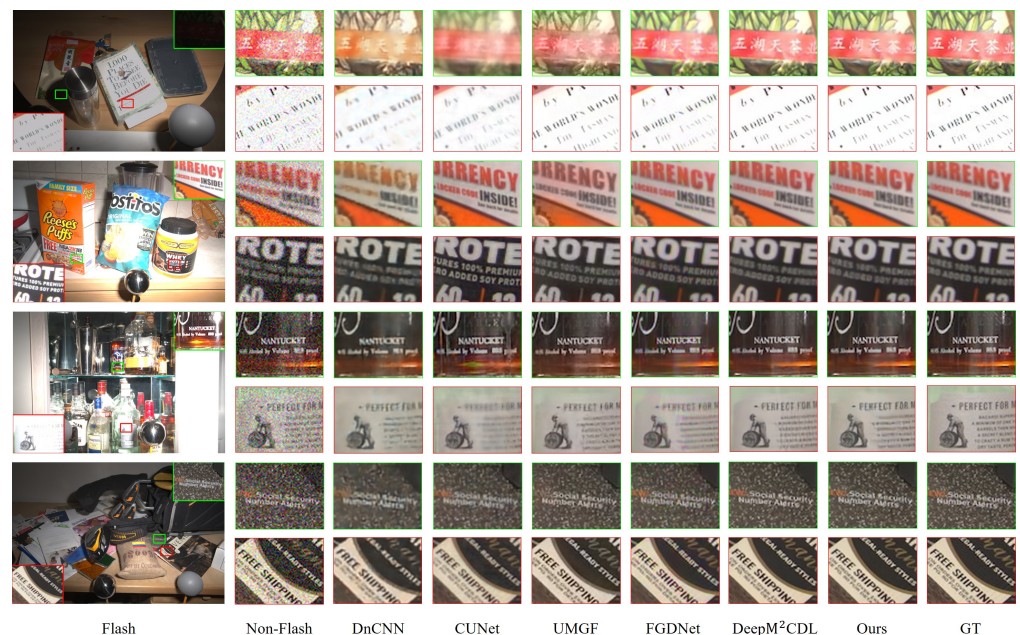

Figure 19: Visual comparison of our method and some representative methods on the MID dataset.

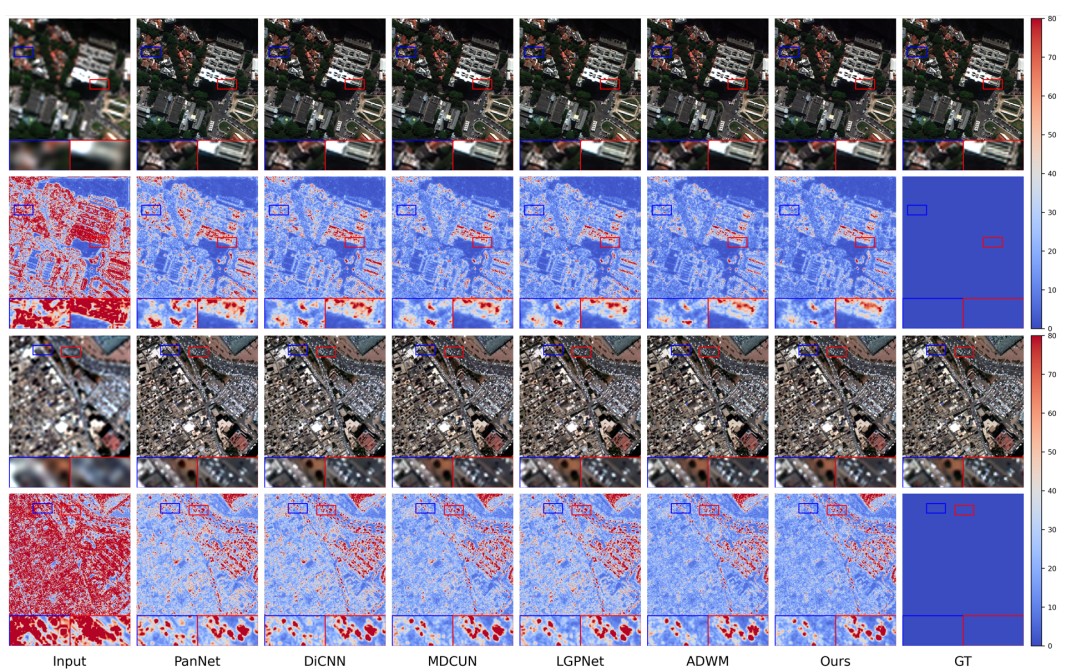

Figure 20: Visual comparison (the first row) and the corresponding error map (the second row) of our method and some representative methods on the WV3 reduced-resolution dataset.

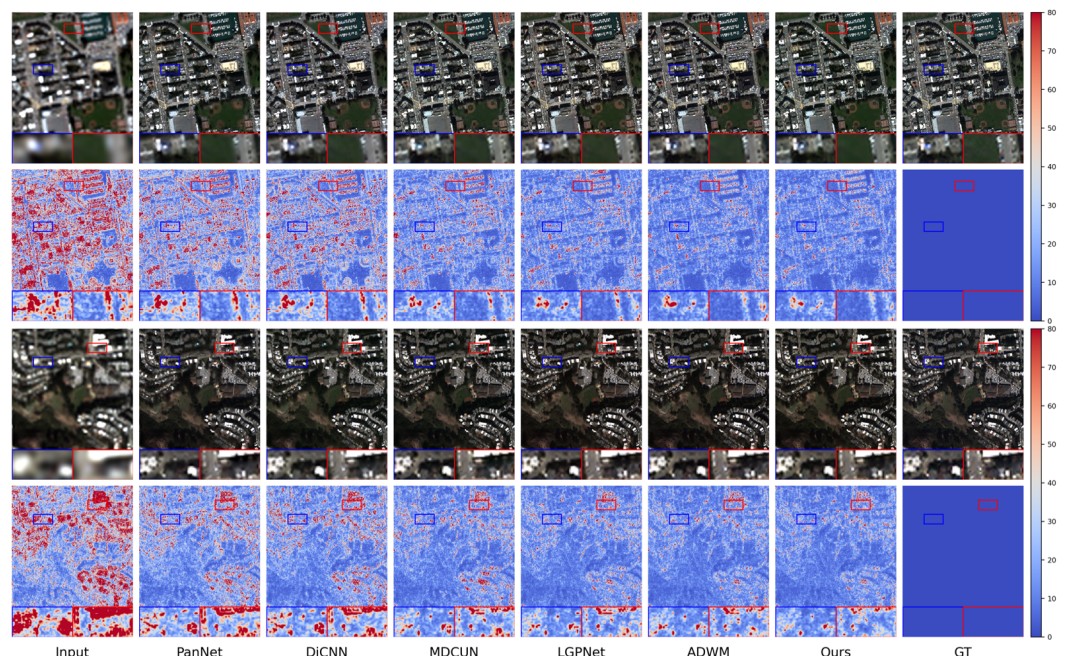

Figure 21: Visual comparison (the first row) and the corresponding error map (the second row) of our method and some representative methods on the QB reduced-resolution dataset.

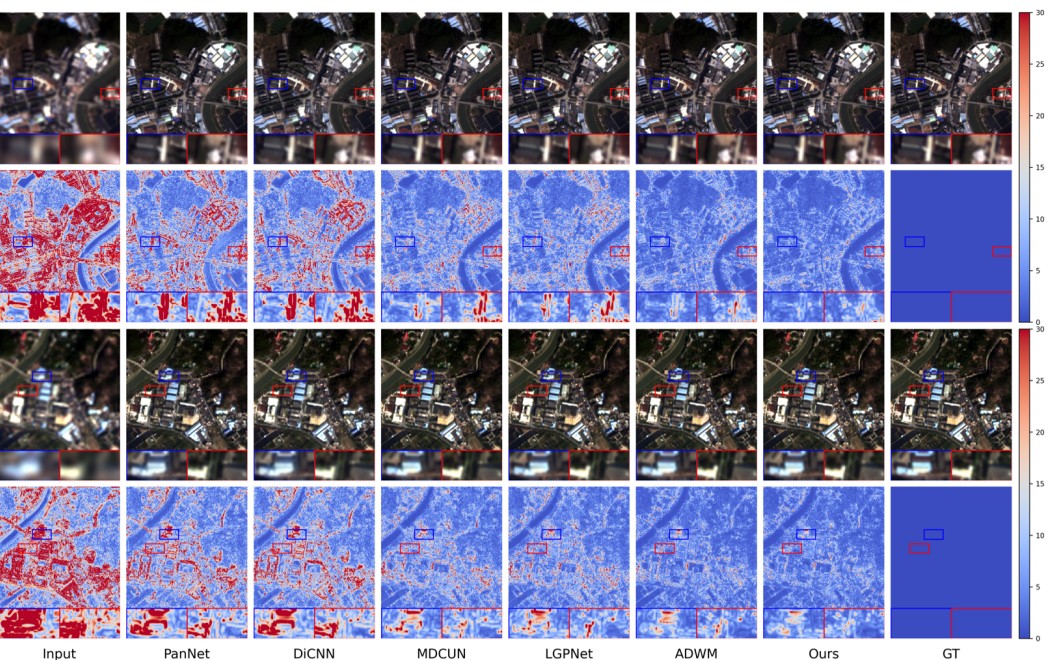

Figure 22: Visual comparison (the first row) and the corresponding error map (the second row) of our method and some representative methods on the GF2 reduced-resolution dataset.

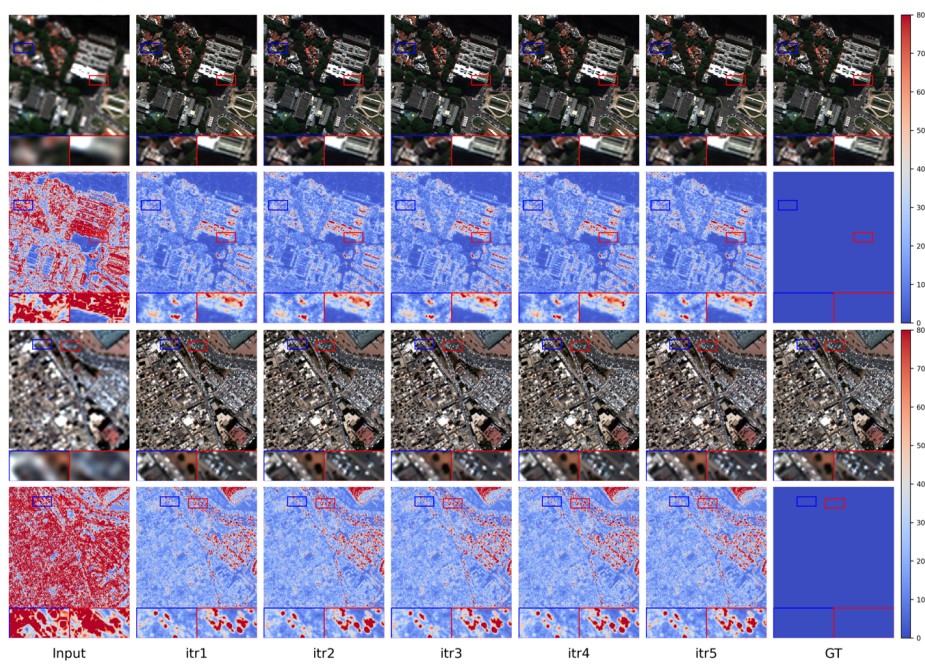

Figure 23: Visual comparison (the first row) and the corresponding error map (the second row) of PGA baseline under different iteration steps on the WV3 reduced-resolution dataset.

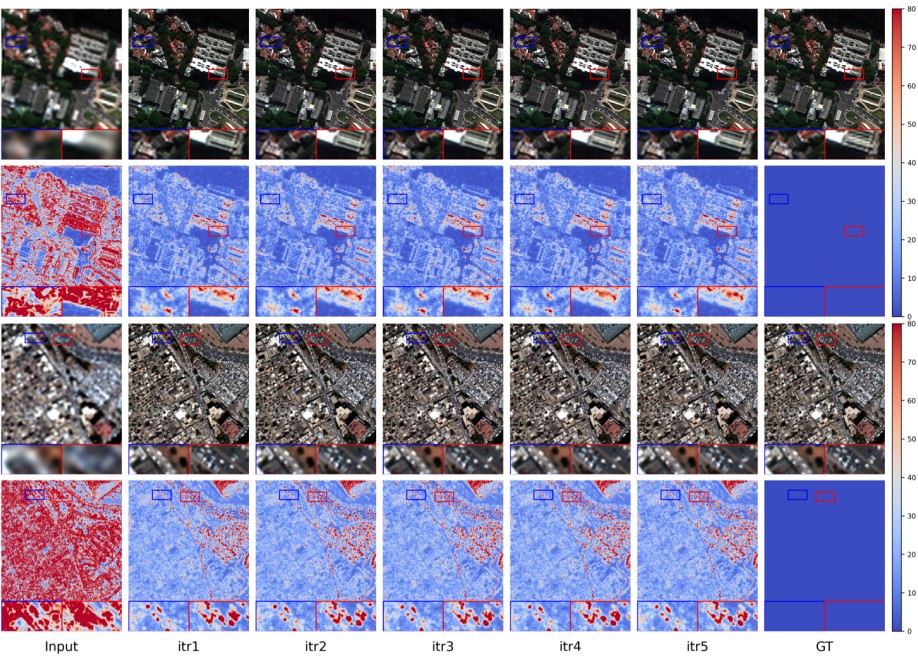

Figure 24: Visual comparison (the first row) and the corresponding error map (the second row) of our SSO-PGA under different iteration steps on the WV3 reduced-resolution dataset.

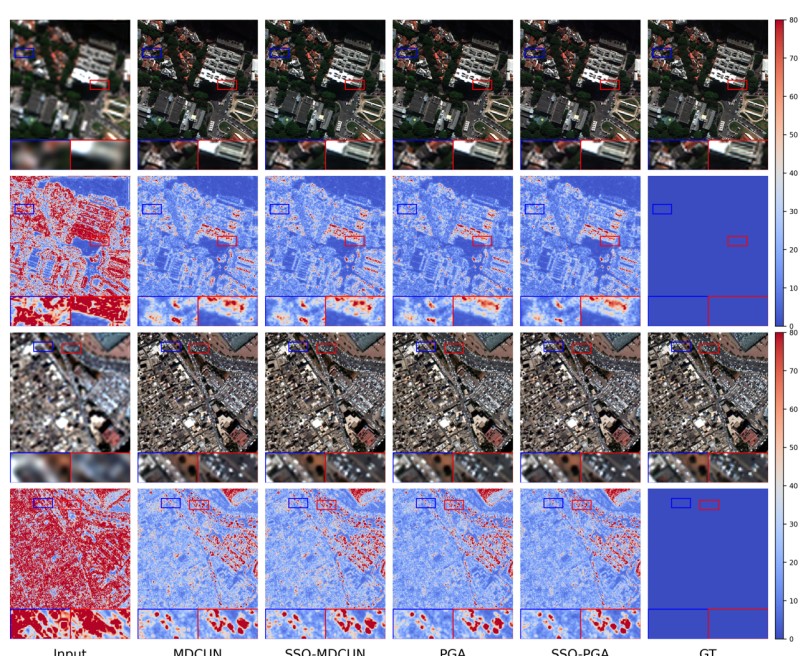

Figure 25: Visual comparison (the first row) and the corresponding error map (the second row) of SSO-PGA vs. PGA baseline, and SSO-MDCUN vs. MDCUN on the WV3 reduced-resolution dataset.

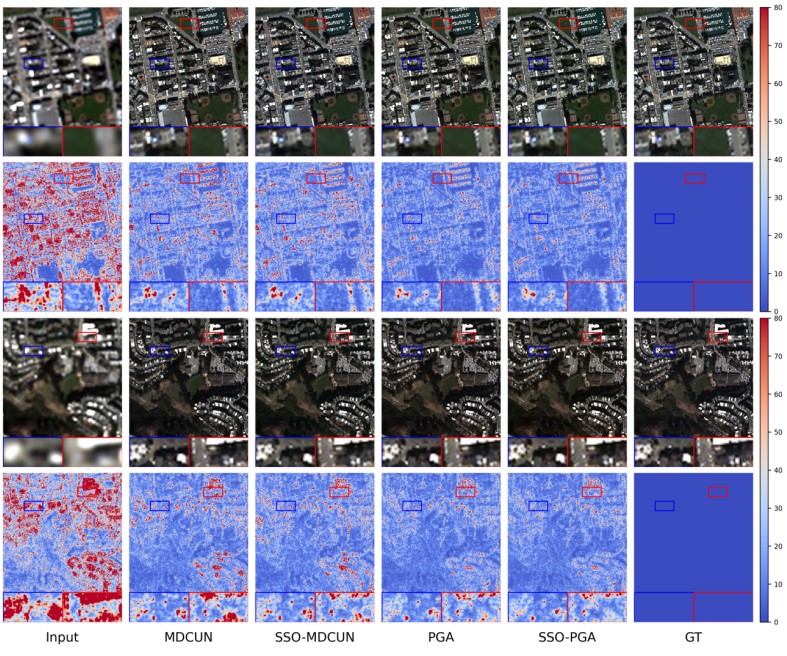

Figure 26: Visual comparison (the first row) and the corresponding error map (the second row) of SSO-PGA vs. PGA baseline, and SSO-MDCUN vs. MDCUN on the QB reduced-resolution dataset.

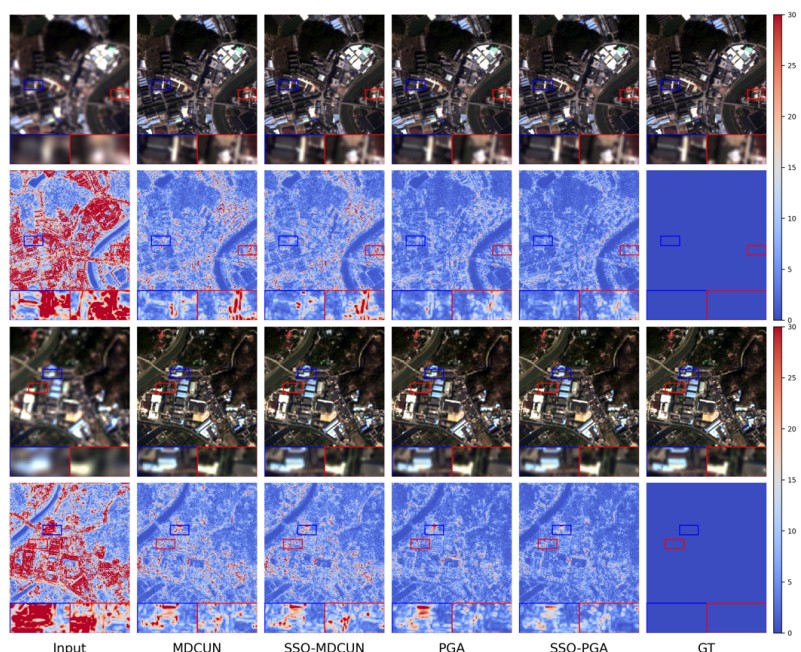

Figure 27: Visual comparison (the first row) and the corresponding error map (the second row) of SSO-PGA vs. PGA baseline, and SSO-MDCUN vs. MDCUN on the GF2 reduced-resolution dataset.

## A.11 THE USE OF LLMS

LLMs did not play a significant role in this research; they were only used for polishing the language and formatting.

