# OpenReview forum: "Robust Non-negative Proximal Gradient Algorithm: Theory and Applications"
_ICLR.cc/2026/Conference — ICLR 2026 Conference Withdrawn Submission_

### Official Review · Reviewer_hNdr · 2025-10-30

**Soundness:** 3
**Presentation:** 2
**Contribution:** 3
**Rating:** 6
**Confidence:** 4

**Summary:**

The authors propose a modification to the Proximal Gradient Algorithm (PGA)'s additive gradient step, with a multiplicative update using a Sliding Sigmoid Operator (SSO). The proposed update maintains the  non negativity and boundedness of an update variable. A sliding parameter controls the effective step size.

**Strengths:**

The novel multiplicative modification to the gradient step, which keeps iterates non negative: This is an interesting idea, with guarantees shown for monotonicity of the error, and equivalent interpretation to traditional gradient descent.

The toy problems set up clearly motivate the intuitions, and the performance is superior to the reported baselines. The proposed method also performs modestly well for practical datasets.

The algorithm is adapted for an unrolled deep neural network to demonstrate its applicability in data-dependent settings.

**Weaknesses:**

1. Some of the claims in the paper are misleading. The broad statement “first to use multiplicative updates for positivity in the PGA context”  is a very strong claim. There have been many multiplicative schemes that attempt to preserve positivity. (Such as Mirror/ exponential descent, projected PG, and re-parametrization updates.) The strong claims could be more justified if they compared against such existing updates upon PG.


2. The authors mention "a straightforward solution" to enforce non-negativity in eq. (6), and build on the intuition for their SSO. It is true that the ratio rule (Lee-Seung 2000) for setting the step size has it's drawbacks. However, there are other fixes (e.g. projected/ proximal methods with indicator-re-parametrization, and mirror/exponential-gradient). Thus it should not be presented as a "straightforward option".


3. The work could be more convincing if they acknowledged these alternative methods, and clearly mention them in the baselines. The extensive results could be included in the supplementary. The authors did include a projection-style baseline (apply ReLU/Softplus after the GD step) and report it underperforms both vanilla PGA and their SSO-PGA, arguing that post-projection “loses information from negative values”. However, this is not sufficient to convince the superiority of SSO over "all" existing gradient step updates, in the context of preserving non-negativity.

**Questions:**

Would appreciate if the authors can address the comments above.

---

### Official Review · Reviewer_qYhz · 2025-10-31

**Soundness:** 3
**Presentation:** 3
**Contribution:** 2
**Rating:** 6
**Confidence:** 4

**Summary:**

This paper proposes a novel proximal gradient algorithm SSO-PGA that replaces the standard additive gradient descent step with a multiplicative update based on a Sliding Sigmoid Operator (SSO). This design enforces non-negativity by construction, improves robustness to hyperparameters, and is embedded into a deep unfolding framework for multi-modal image restoration. The authors provide theoretical convergence guarantees, numerical experiments, and comparisons on inverse problems  (pansharpening, flash-guided denoising), showing consistent improvements over PGA baselines.

**Strengths:**

* The replacement of the standard additive gradient descent step in proximal gradient algorithms (PGA) with a multiplicative update governed by a learnable, bounded Sliding Sigmoid Operator to enforce the non-negativity constraint in the solution is a novel and interesting approach.
* The authors prove that the proposed SSO-PGA iteration is equivalent to a standard gradient step, which motivates its incorporation to deep unfolding deep networks.
* The reported comparisons on Multispectral image Fusion and Flash/Non-Flash denoising show  consistent improvements over standard PGA and various sota methods

**Weaknesses:**

* While this work claims novelty in using a multiplicative update to enforce non-negativity, similar ideas exist in Non-Negative Matrix Factorization (NMF) and EM-type algorithms for Poisson inverse imaging problems, which the authors miss to mention.
* This work compares against several sota methods but ommits comparisons to explicitly non-negative deep networks that utilize projected gradient layers as in Kokkinos and Lefkimmiatis, 2018 and classical constrained optimization solvers that enforce non-negativity via projection (FISTA-based solutions) or barrier methods.
*  In Theorem 2 the condition 0 <= a <= 2/k ||H||^2 -1 is unclear. If  the denominator in the right hand side becomes bigger that 2 the condition is impossible to satisfy.
* The inverse problems studied in this work are rather limited making it unclear whether the proposed approach can show any benefits when applied in other widely studied inverse imaging problems, including demosaicking, supperesolution, deblurring, etc





-- References

* Filippos Kokkinos, Stamatios Lefkimmiatis, "Deep Image Demosaicking using a Cascade of Convolutional Residual Denoising Networks"; Proceedings of the European Conference on Computer Vision (ECCV), 2018, pp. 303-319

**Questions:**

While the authors provide meaningful comparisons with PGA methods, comparisons with other optimization strategies (Majorization-Minimization methods including FISTA) that can successfully enforce the non-negativity constraint in the solution are missing.

---

### Official Review · Reviewer_y36G · 2025-10-31

**Soundness:** 1
**Presentation:** 1
**Contribution:** 1
**Rating:** 0
**Confidence:** 5

**Summary:**

In this work, the authors introduce a modified proximal gradient algorithm (PGA) for non-negative inverse problems, replacing the standard gradient descent step with a learnable sigmoid-based multiplicative operator. The resulting method, termed SSO-PGA, is claimed to inherently enforce non-negativity, improve stability, and offer convergence guarantees. The authors also propose an unfolded deep network version of their algorithm and validate it on multimodal restoration and remote sensing experiments.

While the proposed idea of a learnable multiplicative update is novel and potentially interesting, the paper is fundamentally flawed in its theoretical framing. It contains several misunderstandings of basic convex optimization principles, inaccurate statements about the behavior of standard proximal algorithms, and errors in the provided proofs. As a result, the theoretical contributions cannot be trusted in their current form. On the positive side, the experimental section is clear and the empirical results for remote sensing are promising, but they do not compensate for the conceptual and theoretical issues in the paper.

**Strengths:**

1. The experiment in remote sensing imaging is reasonably well presented and suggest that the proposed method performs competitively.
2. The idea of using a sigmoid-based multiplicative update as a way to enforce non-negativity is original.

**Weaknesses:**

1. The mathematical groundings of the paper contain a large number of errors and inaccuracies.
2. The paper is not well related to current works. This makes it difficult to assess its novelty.

**Questions:**

**Major comments**
1. The abstract and introduction make factually incorrect claims such as “PGA often yields unstable convergence.” This is misleading at best: proximal gradient algorithms are among the most well-established optimization methods, with strong convergence guarantees in convex and even certain non-convex settings. If the authors wish to discuss instability, they should clearly define what kind of “instability” they refer to (e.g., sensitivity to step size, oscillations in non-convex cases), and support this with references or evidence.
2. The literature review omits essential prior work on replacing proximal operators with learned or non-standard mappings, notably: Plug-and-Play Priors (Venkatakrishnan et al., 2013), Meinhardt et al., Learning Proximal Operators (CVPR 2017), Ryu et al., "Plug-and-Play Methods Provably Converge with Properly Trained Denoisers", Pesquet et al., "Learning Maximally Monotone Operators for Image Recovery". These works provide much more rigorous frameworks for combining learned components with optimization steps, and the authors should position their approach relative to them.
3. The claim that “the update rule in Eq. (4) may yield negative values” is incorrect. Indeed, if $g$ contains the non-negative orthant $g = \iota_C$ with $C=\mathbb{R}_+$, the proximal operator is simply the projection on $C$ which enforces non-negativity automatically. If $g$ is the sum of $\iota_C$ and another convex function, although there is no closed-form in general, the prox will be positive valued (elementwise). There is no need to modify the gradient step.
4. The proof of Theorem 1 is invalid. The value $\rho_i$ depends on $y^{t-1}$, creating a circular argument. $\exists \rho_i$ assumes that $\rho_i$ is independant of $y^{t-1}$. Otherwise, this creates a circular dependency (or the functional $\nabla E$ needs to be changed). Furthermore, the derivation confuses scalar and vector quantities: gradients of real-valued functions are vectors, so the equality proposed only holds pointwise and this should be specified by the authors.
5. Theorem 2: The argument that the algorithm “converges to a local minimum” because a sequence is non-increasing and bounded below is incorrect. A non-decreasing bounded sequence may not converge to zero; the authors need to establish that the gradient norm tends to zero, and then resort to analysis arguments for completing the proof. But as is, the authors just show that the cost function is upper-bounded, which is far from a convergence result.
6. Several lemmas are standard results from convex analysis and add little value in their current form without clear adaptation to the proposed method. Their proof should be ommited. The text also conflates the inverse problem with the data-fidelity term (e.g., line 213). This distinction should be clarified.
7. The proposed “stability” argument is not grounded in standard optimization theory. PGA already guarantees stable and monotone convergence under mild assumptions. If the authors wish to argue for robustness to step size or noise, this must be formalized and supported by analysis or empirical ablation.
8. The notation $f^*$ in line 275 appears to denote a Jacobian transpose, which should be clarified.
9. Problem (24) is trivial: it has a closed-form solution when H is the identity, so it does not meaningfully demonstrate algorithmic benefits.
10. Figure 6 should include a convergence plot over many more iterations (e.g., 100+) and track the evolution of the objective function or iterate norm, as is standard in optimization papers (e.g., Ryu et al., Pesquet et al.).

**Minor comments**

1. Notations are inconsistent throughout. The authors have introduced $f$ in the beginning but switch for $\mathcal{E}$ in the middle.
2. Did the authors expeirment plug-and-play algorithms or diffusion for the considered experiments? I think this would potentially be good candidates for this problem.

---

### Official Review · Reviewer_Sr9p · 2025-11-01

**Soundness:** 3
**Presentation:** 2
**Contribution:** 2
**Rating:** 4
**Confidence:** 4

**Summary:**

This paper proposes a novel proximal gradient algorithm called SSO-PGA, which replaces the standard additive gradient descent step with a multiplicative update rule using a Sliding Sigmoid Operator (SSO). The authors claim that this modification inherently enforces non-negativity constraints, improves stability, and enhances convergence behavior. The method is further unfolded into a deep network and evaluated on two image fusion tasks: multispectral image fusion and flash-guided non-flash denoising. Experimental results show that SSO-PGA outperforms traditional PGA and several state-of-the-art methods in terms of both quantitative metrics and visual quality.

**Strengths:**

The proposed multiplicative update rule based on the Sliding Sigmoid Operator is novel and offers an interesting alternative to traditional gradient descent.

**Weaknesses:**

- The motivation for using a multiplicative update to enforce non-negativity is weak. Non-negativity can be easily handled in traditional optimization via projection or by including a non-negativity constraint in the proximal operator. The authors did not compare SSO-PGA against such standard constrained optimization baselines.

- When the optimization algorithm is unfolded into a deep network, the original properties of the iterative algorithm (e.g., convergence guarantees) may not be preserved. In such cases, non-negativity can be trivially enforced using activation functions (e.g., ReLU or Softplus), which the authors briefly compared but did not thoroughly justify why SSO is fundamentally better in the deep learning context.

- The title and claims of the paper are overly broad. The method is only validated on image fusion tasks, yet the title and abstract suggest a general-purpose optimization improvement. This overstates the contribution and applicability.

- The explanation of why SSO-PGA performs better—beyond non-negativity—is insufficient. The improvement may stem from other factors such as adaptive step sizes or better gradient conditioning, which are not sufficiently analyzed.

**Questions:**

See Weaknesses

---

### Note · Authors · 2025-11-19

I have read and agree with the venue's withdrawal policy on behalf of myself and my co-authors.